# Male rodent perirhinal cortex, but not ventral hippocampus, inhibition induces approach bias under object-based approach-avoidance conflict

Sandeep S Dhawan[1], Carl Pinter[1], Andy CH Lee[1,2]*, Rutsuko Ito[1,3]*

[1]Department of Psychology (Scarborough), University of Toronto, Toronto, Canada; [2]Rotman Research Institute, Baycrest Centre, Toronto, Canada; [3]Department of Cell and Systems Biology, University of Toronto, Toronto, Canada

**Abstract** Neural models of approach-avoidance (AA) conflict behavior and its dysfunction have focused traditionally on the hippocampus, with the assumption that this medial temporal lobe (MTL) structure plays a ubiquitous role in arbitrating AA conflict. We challenge this perspective by using three different AA behavioral tasks in conjunction with optogenetics, to demonstrate that a neighboring region in male rats, perirhinal cortex, is also critically involved but only when conflicting motivational values are associated with objects and not contextual information. The ventral hippocampus, in contrast, was found not to be essential for object-associated AA conflict, suggesting its preferential involvement in context-associated conflict. We propose that stimulus type can impact MTL involvement during AA conflict and that a more nuanced understanding of MTL contributions to impaired AA behavior (e.g., anxiety) is required. These findings serve to expand upon the established functions of the perirhinal cortex while concurrently presenting innovative behavioral paradigms that permit the assessment of different facets of AA conflict behavior.

*For correspondence:
andych.lee@utoronto.ca (ACHL);
rutsuko.ito@utoronto.ca (RI)

**Competing interest:** The authors declare that no competing interests exist.

## Editor's evaluation

In this important study the authors combined innovative object-based conflict assays with optogenetic silencing to probe the role of the perirhinal cortex in motivational conflict. The evidence provided by the authors is convincing and provides new insight into how conflicting motivation is processed. This paper will interest neuroscientists studying the neuronal mechanisms underlying approach-avoidance decisions.

## Introduction

Approach-avoidance (AA) conflict is elicited when an organism experiences competition between incompatible motivations of being attracted to and repelled by the same goal stimulus. Its effective resolution is vital for survival and everyday decision-making, while its dysfunction manifests across a spectrum of psychiatric disorders including anxiety and addiction (*Aupperle and Paulus, 2010*; *Fricke and Vogel, 2020*). Since the septo-hippocampal system was first theorized to mediate a behavioral inhibition system (BIS) that is activated to suppress approach responses under conflict situations (*Gray and McNaughton, 2000*), converging cross-species empirical evidence has highlighted a critical role for the ventral (rodent) or anterior (primate), but not dorsal/posterior portion of the hippocampus (HPC), in modulating anxiety and AA responding during high motivational conflict (*Ito and Lee, 2016*). For example, in addition to increasing anxiolytic behavior in rodents, lesion or

pharmacological inactivation of the ventral HPC (vHPC) increases approach behaviors to motivationally conflicting learned stimuli (*Bannerman et al., 2014*; *Bannerman et al., 2003*; *Bannerman et al., 2002*; *Schumacher et al., 2018*; *Schumacher et al., 2016*; *Yeates et al., 2020*). Similarly, human functional magnetic resonance imaging (fMRI) and patient neuropsychological studies with analogous AA tasks have revealed anterior HPC involvement when participants experience high AA conflict (*Bach et al., 2014*; *O'Neil et al., 2015*).

Given the focus on the role of the HPC in AA conflict processing, very limited empirical work has explored the potential contributions of the surrounding medial temporal lobe (MTL) cortices. This is somewhat surprising given the intricate anatomical and functional relationships between MTL structures. Moreover, a revised formulation of the BIS postulated the involvement of the entorhinal cortex and perirhinal cortex (PRC) in the detection and resolution of AA conflict (*Gray and McNaughton, 2000*), a suggestion that has not yet, to our knowledge, been fully examined empirically. Theoretical models of MTL function posit that stimulus type can impact MTL structure recruitment during cognition. Specifically, the HPC is implicated in scene and context-based processing, while the PRC is predominantly involved in object-associated processes, both within the domain of memory and even beyond (*Graham et al., 2010*; *Murray et al., 2007*; *Zeidman and Maguire, 2016*). Since ethological anxiety tasks and paradigms of AA conflict processing have typically employed spatial stimuli (e.g., scene images) or required environmental exploration (e.g., maze navigation) (*Bach et al., 2014*; *Bannerman et al., 2002*; *O'Neil et al., 2015*; *Schumacher et al., 2018*; *Schumacher et al., 2016*), it raises the question of whether the HPC plays a ubiquitous role in AA conflict processing across all stimulus domains or whether other MTL structures may also play a critical role depending on the stimuli involved.

Here, we designed two rodent AA tasks using objects as target stimuli to test the hypothesis that differential MTL recruitment occurs during AA conflict processing in a stimulus-type specific manner in rodents. In both tasks, rats were trained to associate object pairs with either appetitive or aversive outcomes and were subsequently presented with these objects in a conflict arrangement (simultaneous presentation of appetitive and aversive objects) while the PRC or vHPC (CA3 subfield) was optogenetically inhibited. However, each task was conducted in a different apparatus, specifically a runway or shuttle box, to elicit a different type of avoidance behavior (passive vs. active). The PRC was also inhibited while animals completed an established 'contextual' AA task, known to be dependent on the vHPC (*Schumacher et al., 2018*; *Schumacher et al., 2016*; *Yeates et al., 2020*). Across both object-based tasks, we observed that inhibition of rodent PRC, but not the vHPC, induced a significant approach bias in response to objects signaling availability of both reward and punishment. In contrast, there was no impact of PRC inhibition when conflict was elicited by contextual cues. These findings implicate a hitherto unconsidered substrate in the arbitration of AA conflict; they emphasize the need to consider the involvement of the broader MTL in AA conflict processing and have implications for our understanding of the neural substrates underlying disorders in which AA conflict behavior is compromised.

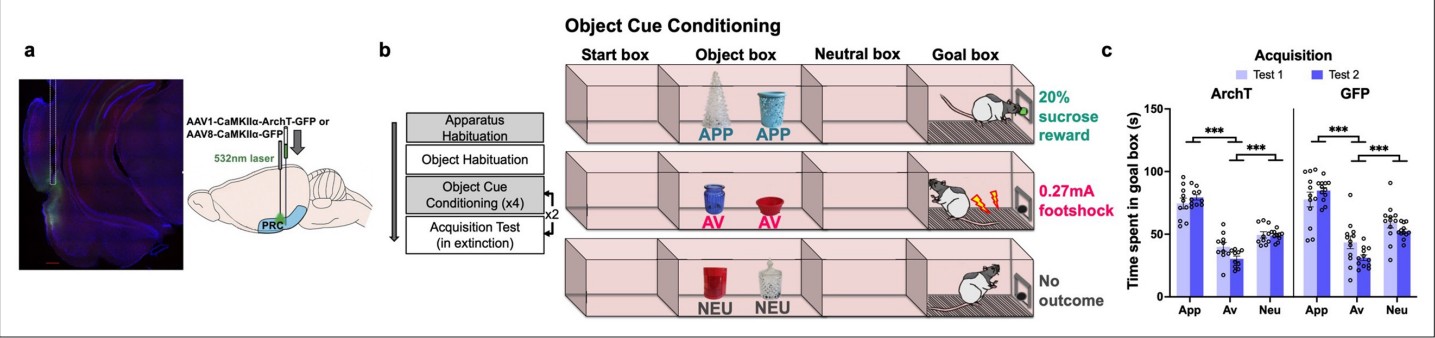

**Figure 1.** Object approach-avoidance (AA) conflict runway task. Rats expressing AAV1-CaMKIIa-ArchT-GFP (archaerhodopsin T [ArchT]) or AAV8-CAMKIIa-GFP (green fluorescent protein [GFP]) in the perirhinal cortex (PRC, **a**) underwent object cue conditioning to learn the outcomes associated with appetitive (APP), aversive (AV), and neutral (NEU) object pairs (**b**). Three-way ANOVA of acquisition data (mean ± SEM) indicated successful learning in both ArchT and GFP groups (**c**). Post-hoc tests with Bonferroni correction applied. ***p<0.001.

## Results

### Optogenetic inhibition of PRC increases approach behavior to motivational conflict represented by objects in a runway paradigm

Rats were first surgically infused with inhibitory archaerhodopsin T (ArchT) or green fluorescent protein only (GFP) and implanted bilaterally with optic fibers into the PRC (*Figure 1a*). Rats were then trained on a novel task in which they first acquired the incentive values of three pairs of objects (appetitive [sucrose], aversive [electric shock], and neutral [no outcome] pairs) in a customized runway apparatus comprising four successive compartments: a start box; an object box containing two objects; a neutral box in which the rat was held temporarily; and finally, a goal box, in which the associated outcomes were delivered during training (*Figure 1b*). Objects were selected to be visually and textually distinct from one another since PRC is implicated in the discrimination of objects with overlapping features (*Murray et al., 2007*).

Acquisition of object valences was assessed after four (test 1) and eight (test 2) conditioning sessions, without laser treatment. PRC groups acquired the object-outcome associations successfully by test 2, with rats spending the most time in the goal box after exposure to the appetitive object pair (p<0.001), and the least time after aversive object pair exposure (p<0.001), compared to goal box time in neutral trials (valence: $F_{(2,40)}$=396.99, p<0.001, $\eta_p^2 = 0.95$). Rats also exhibited a significant reduction in goal box time during aversive test session between the two acquisition tests (valence × test: $F_{(2,40)}$=16.86, p<0.001, $\eta_p^2 = 0.46$) (*Figure 1c*). The pattern of valence acquisition was comparable between ArchT and GFP animals (construct: $F_{(1,20)}$=1.51, p=0.23, $\eta_p^2 = 0.07$; valence × construct: $F_{(2,40)}$=1.1, p=0.34, $\eta_p^2 = 0.05$; test × construct: $F_{(2,40)}$=0.2, p=0.66, $\eta_p^2 = 0.01$).

Animals were then administered a series of tests in which recombinations of the learned objects were presented to elicit a high (appetitive-aversive) or low (appetitive-neutral; aversive-neutral) level of motivational conflict, or no conflict (neutral object pair was presented) (*Figure 2a*). When animals were exposed to a high conflict object pair, optogenetic inhibition of PRC (laser on) significantly increased time spent in the goal box compared with trials completed without inhibition (laser off), and compared to GFP control animals with laser on and off (laser × construct: $F_{(1,20)}$=40.73, p<0.001, $\eta_p^2 = 0.67$; all post hoc: p<0.001) (*Figure 2b*), indicative of an increase in approach behavior under motivational conflict. An analysis of the 'difference score' between the two conflict test sessions (i.e., laser on–laser off) also indicated that laser-treated ArchT animals spent more time in the goal box than GFP animals (ArchT: 29.72 (M)±14.23 (SD), GFP: –1.41±8.37, t(20) = 6.38, p<0.001). Furthermore, PRC inhibition led to a significant decrease in the number of retreats from the goal box (laser × construct: $F_{(1,20)}$=24.08, p<0.001, $\eta_p^2 = 0.55$), although there was no effect on the number of goal box entries (laser × construct: $F_{(1,20)} = 3.12$, p=0.09, $\eta_p^2 = 0.14$) (*Figure 2c*) during the choice period. PRC inhibition also had no impact on the latency to enter (LTE) each of the object, neutral and goal boxes (laser × construct: $F_{(1,20)} = 2.24$, p=0.15, $\eta_p^2 = 0.1$; laser × construct × box: $F_{(2,40)}$=0.47, p=0.63, $\eta_p^2 = 0.02$), nor did it have an effect on the exploration of appetitive and aversive objects (laser × construct: $F_{(1,20)} = 3.10$, p=0.09, $\eta_p^2 = 0.13$; laser × construct × object: $F_{(1,20)}$=0.15, p=0.70, $\eta_p^2 = 0.007$) (*Figure 2—figure supplement 1a–b*). Collectively, these results indicate that when faced with high motivational conflict, PRC-inhibited rats entered the goal box as readily as control animals but stayed longer and retreated less, indicative of a potentiated approach bias under conflict. This effect was not observed in the neutral object test: PRC inhibition had no effect on the time spent in the goal box after exploring neutral objects (laser × construct: $F_{(1,18)}$=1.02, p=0.33, $\eta_p^2 = 0.05$) (*Figure 2d*).

When PRC-inhibited animals were exposed to 'low conflict' object pairs (appetitive-neutral and aversive-neutral), their time in the goal box increased (laser × construct: $F_{(1,16)}$=9.37, p=0.007, $\eta_p^2 = 0.37$) (*Figure 2e*), but there was no corresponding increase in the number of entries into (laser × construct: $F_{(1,16)}$=0.47, p=0.50, $\eta_p^2 = 0.02$) or retreats from (laser × construct: $F_{(1,16)}$=0.54, p=0.47, $\eta_p^2 = 0.04$) the goal box during the choice period (*Figure 2f–g*). An analysis of the difference score (laser on–laser off) for the two low conflict tests revealed that laser treatment increased goal box time for ArchT animals compared with GFP, when faced with both App-Neu (ArchT: 22.13 (M)±21.61 (SD), GFP: 1.81±7.49, Mann-Whitney U=15, p=0.027, two-tailed) and Av-Neu (ArchT: 13.32±8.45, GFP: 1.07±10.77, t(16) = 2.71, p=0.016) pairings. PRC inhibition had no impact on the LTE into any of the runway boxes (laser × construct: $F_{(1,16)} = 0.24$, p=0.63, $\eta_p^2 = 0.02$; laser × construct × box: $F_{(2,32)}$=0.21, p=0.81, $\eta_p^2 = 0.01$), or valence of the object pairing (valence: $F_{(1,16)} = 2.38$, p=0.15, $\eta_p^2 = 0.17$) (*Figure 2—figure supplement 1c–d*). There was no effect of PRC inhibition on the

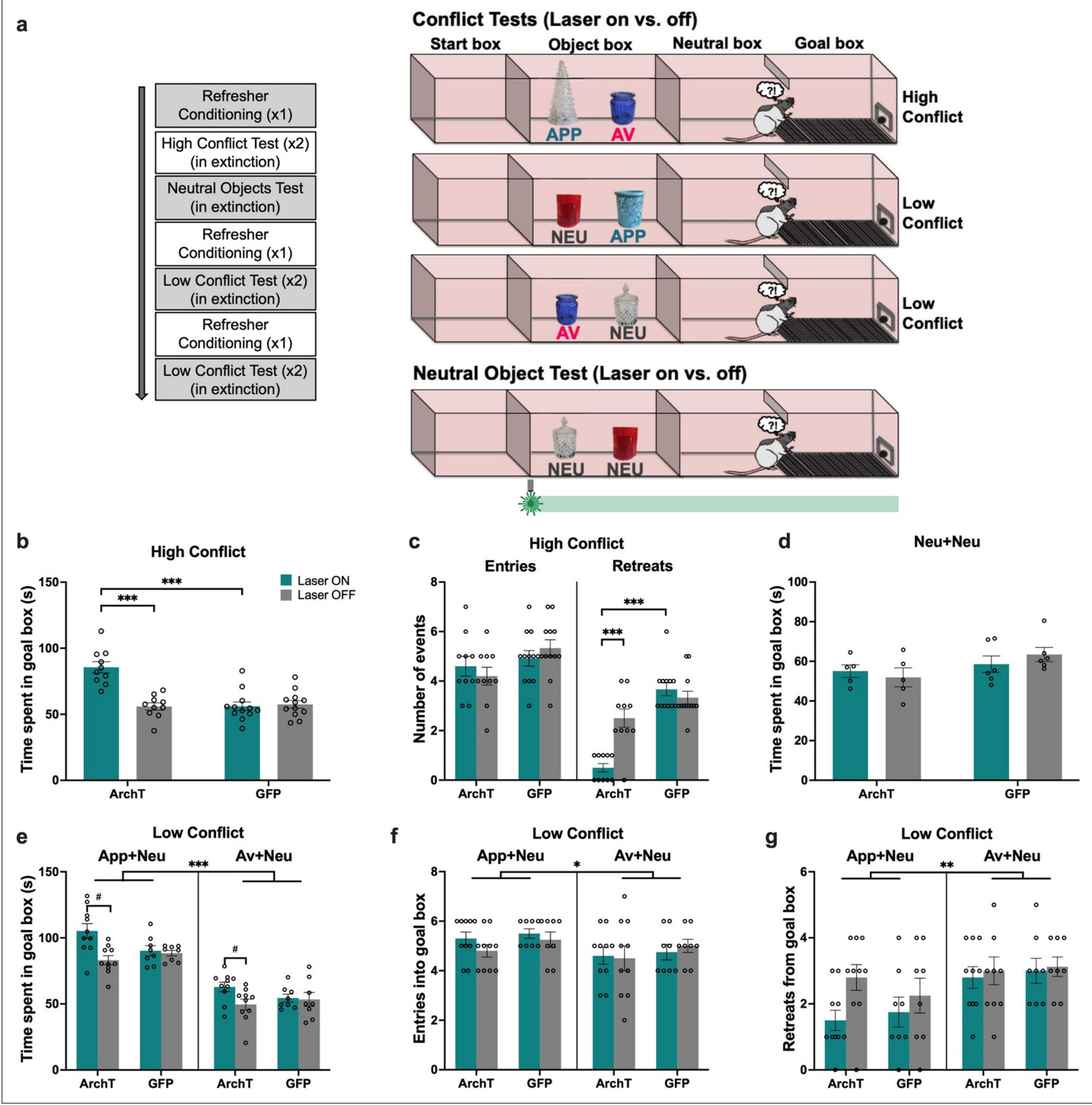

**Figure 2.** Impact of perirhinal cortex (PRC) inhibition on object approach-avoidance (AA) conflict runway task performance. (**a**) Rats (n=10 archaerhodopsin T [ArchT]; n=12 green fluorescent protein [GFP]) underwent a series of tests conducted in extinction: high conflict (APP+AV objects), neutral (NEU+NEU), and low conflict (APP+NEU or AV+NEU). (**b–c**) PRC inhibition (laser ON during the entire session) significantly increased time spent in the goal box and reduced the number of retreats in the high conflict test. (**d**) There was no effect of PRC inhibition on AA behavior in the neutral test. (**e**) Similar to the high conflict test, PRC inhibition increased time spent in the goal box in both low conflict tests. (**f–g**) PRC inhibition did not impact the number of entries or retreats in the App+Neu or Av+Neu low conflict tests, although there was a main effect of valence, with a greater number of entries for App+Neu and more retreats for Av+Neu. All figures show mean values ± SEM. Three-way ANOVA was conducted for data shown in c,e-g, and two-way ANOVA was conducted for data in b,d. Post-hoc tests with Bonferroni correction were conducted to further investigate significant interactions in all datasets except in d.***p<0.001, **p<0.01, *p<0.05.

*Figure 2 continued on next page*

Figure 2 continued

The online version of this article includes the following figure supplement(s) for figure 2:

**Figure supplement 1.** Additional perirhinal cortex (PRC) rat data for the approach-avoidance (AA) conflict runway task.

exploration of the App-Neu and Av-Neu object pairings (laser × construct: $F_{(1,16)}=1.14$, p=0.30, $\eta_p^2$ = 0.07; laser × construct × object: $F_{(1,16)}=0.02$, p=0.88, $\eta_p^2$ = 0.001), with comparable object exploration between valence pairings (valence: $F_{(1,16)} = 1.64$, p=0.22, $\eta_p^2$ = 0.09; valence × construct: $F_{(1,16)}=3.73$, p=0.07, $\eta_p^2$ = 0.19) (*Figure 2—figure supplement 1e–f*).

Notably, PRC inhibition did not impair the ability to discriminate appetitive and aversive valences (valence: $F_{(1,16)}=229.64$, p<0.001, $\eta_p^2$ = 0.94; laser × valence × construct: $F_{(1,16)}=1.14$, p=0.3, $\eta_p^2$ = 0.07), with all animals spending significantly more time in the goal box for appetitive-neutral pairings than aversive-neutral (p<0.001, *Figure 2e*). Furthermore, all PRC rats made fewer entries (p=0.039) and exhibited more retreat behavior (p<0.0001) when presented with an aversive-neutral

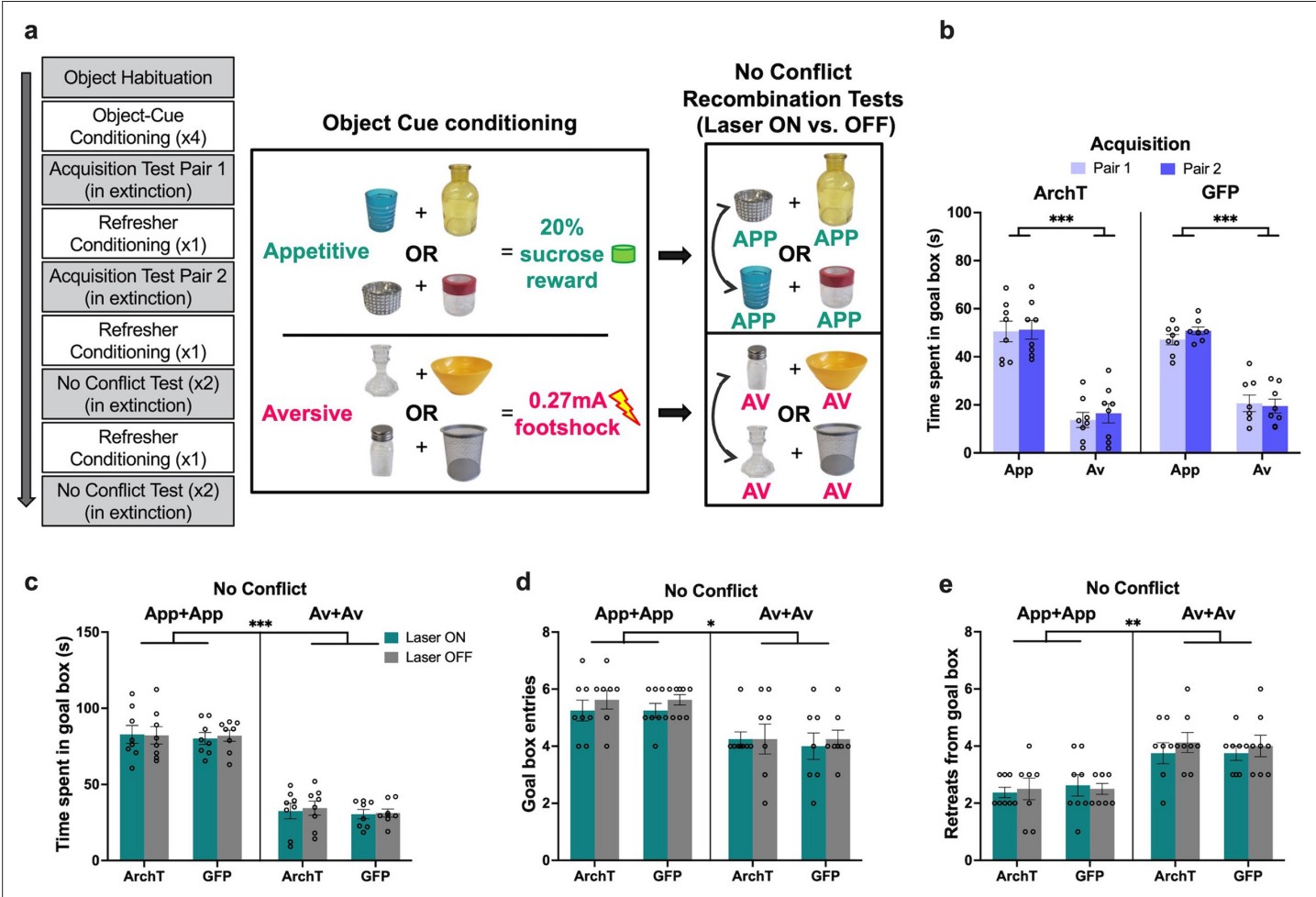

**Figure 3.** Impact of perirhinal cortex (PRC) inhibition on 'no conflict' approach-avoidance (AA) conflict runway task performance. (**a**) A subset of rats (n=8 archaerhodopsin T [ArchT]; n=8 green fluorescent protein [GFP]) underwent a no conflict recombination task, in which they first learned a new set of appetitive (APP) or aversive (AV) object pairs and were then presented with recombined object pairs composed of objects of the same valence. (**b**) Both ArchT and GFP PRC rats successfully learned a new set of APP and AV object pairs for the no conflict recombination tests. (**c–e**) PRC inhibition did not affect AA behavior on the no conflict tests. A main effect of valence was observed for all measures, indicating intact valence retrieval for the recombined test pairs. All figures show mean values ± SEM. Three-way ANOVA followed by post-hoc tests with Bonferroni corrections was conducted for all data shown. ***p<0.001, **p<0.01, *p<0.05.

The online version of this article includes the following figure supplement(s) for figure 3:

**Figure supplement 1.** Additional perirhinal cortex (PRC) rat data for the approach-avoidance (AA) no conflict recombination task.

pairing compared with an appetitive-neutral pairing (*Figure 2f–g*), indicating that valence recall was intact in PRC-inhibited rats.

To rule out the possibility that the observed effects of PRC inhibition were driven by a failure to respond to novel recombinations of object pairings rather than conflict processing, we repeated the runway task in which animals were trained to associate a new set of four object pairs with either appetitive or aversive outcomes (two pairs each) (*Figure 3a*). Animals acquired the cue-outcome associations successfully by the first test, and spent significantly more time in the goal box after exposure to the appetitive object pairs compared with the aversive pairs, with comparable acquisition behavior between sets of objects pairs within each valence (valence: $F(1,14)=259.27$, $p<0.001$, $\eta_p^2 = 0.95$; pairing: $F(1,14)=1.67$, $p=0.22$, $\eta_p^2 = 0.11$; valence × pairing: $F(1,14)=0.27$, $p=0.61$, $\eta_p^2 = 0.02$; *Figure 3b*). The pattern of object cue acquisition was comparable between ArchT and GFP animals (construct: $F(1,14)=0.16$, $p=0.69$, $\eta_p^2 = 0.01$; pairing × construct: $F(1,14)=0.04$, $p=0.86$, $\eta_p^2 = 0.003$).

Rats were then administered a within-valence 'no conflict' recombination test (appetitive-appetitive; aversive-aversive). PRC inhibition did not lead to significant changes in the time spent in the goal box after exposure to novel appetitive or aversive object pairs (laser: $F(1,14)=0.10$, $p=0.76$, $\eta_p^2 = 0.007$; laser × construct: $F(1,14)=0.01$, $p=0.92$, $\eta_p^2 = 0.001$) (*Figure 3c*). An analysis of the difference score (laser on–laser off) for the two respective no conflict tests revealed that laser treatment had no effect on goal box time when animals encountered either App-App (ArchT: 0.65 (M)±9.27 (SD), GFP: –1.81±16.33, $t(14)=0.37$, $p=0.72$) or Av-Av (ArchT: –1.97 (M)±19.80 (SD), GFP: –0.72±12.54, $t(14) = –0.15$, $p<0.88$) pairings. Furthermore, both ArchT and GFP animals readily discriminated between valences during this test (valence: $F(1,14) = 371.95$, $p<0.001$, $\eta_p^2 = 0.96$; valence × construct: $F(1,14)=0.05$, $p=0.82$, $\eta_p^2 = 0.004$). PRC inhibition also had no effect on the number of entries into (laser × construct: $F(1,14)=0.06$, $p=0.82$, $\eta_p^2 = 0.004$) nor retreats from the goal box (laser × construct: $F(1,14)=0.15$, $p=0.71$, $\eta_p^2 = 0.01$), and furthermore, all PRC-inhibited rats made fewer

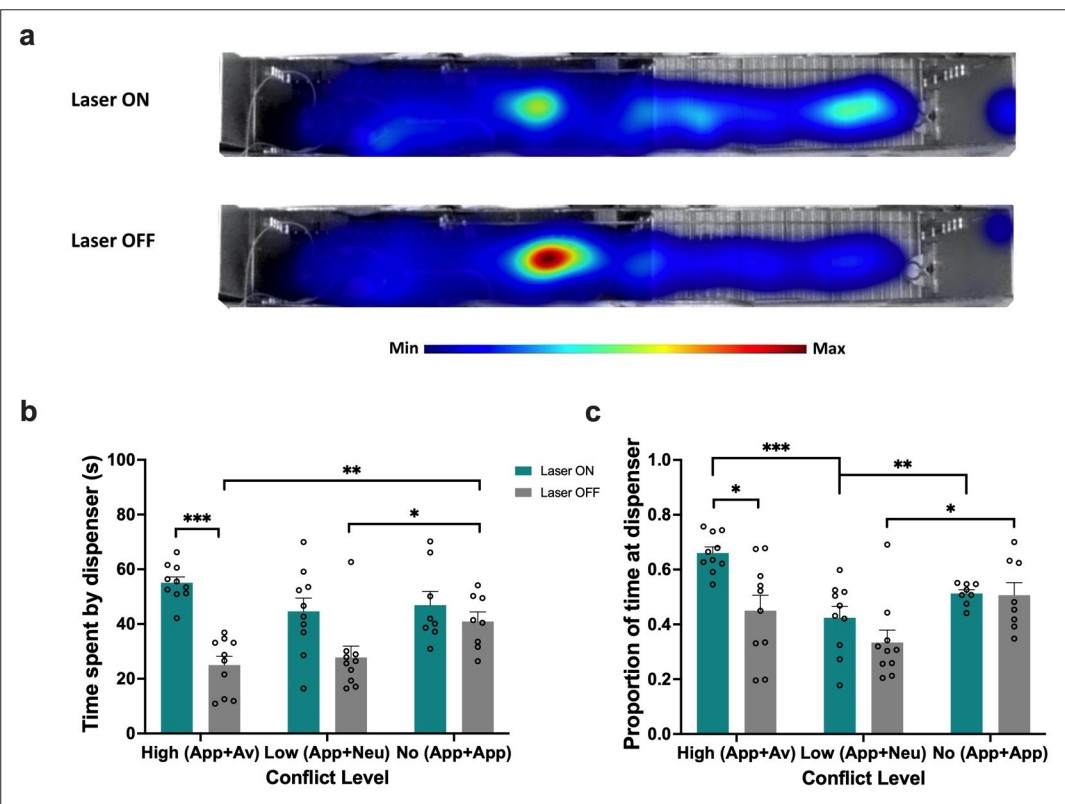

**Figure 4.** Perirhinal cortex (PRC) inhibition increases reward location approach behavior. (**a**) Heatmap plots for the high conflict test. (**b–c**) PRC-inhibited rats spent more time by the sucrose dispenser, as measured by total time or proportion of time, in the high and low conflict tests but not in the no conflict test. All figures show mean values ± SEM. Two-way ANOVA, followed by post-hoc tests with Bonferroni correction was conducted for all data shown. \*\*\*p<0.001, \*\*p<0.01, \*p<0.05.

entries (p<0.001) and exhibited more retreat behavior (p<0.0001) when presented with an aversive pairing compared with an appetitive pairing (*Figure 3d–e*). PRC inhibition also had no impact on the LTE data into any of the runway boxes (laser × construct: F(1,14)=1.7, p=0.21, $\eta_p^2 = 0.11$; laser × construct × box: F(2,28)=0.30, p=0.74, $\eta_p^2 = 0.02$) regardless of the valence of the object pairing (valence: F(1,14)=2.99, p=0.11, $\eta_p^2 = 0.18$); nor did it have an effect on the exploration of appetitive and aversive object pairs (laser × construct: F(1,14)=1.07, p=0.32, $\eta_p^2 = 0.07$; laser × construct × object: F(1,14)=0.07, p=0.80, $\eta_p^2 = 0.005$), with comparable object exploration between valence pairings (valence: F(1,14)=3.67, p=0.08, $\eta_p^2 = 0.21$; valence × construct: F(1,14)=0.39, p=0.55, $\eta_p^2 = 0.03$; *Figure 3—figure supplement 1a–d*).

Finally, to investigate whether animals exhibiting approach bias spent an increased amount of time in the vicinity of the sucrose dispenser in expectation of reward (i.e., reward location approach), we analyzed the amount of time and proportion of total goal box time that ArchT animals spent in a demarcated 'dispenser zone' (final 18 cm or 1/3 of goal box) during the high (App/Av), low (App/Neu), and no conflict (App/App) test sessions. When faced with high conflict, PRC inhibition led to animals spending significantly more time by the sucrose dispenser compared to high conflict test sessions without optogenetic inhibition (p<0.0001), whereas an increase in time spent by the dispenser elicited by low conflict stimuli approached significance (p=0.052; laser: F(1,18)=24.34, p<0.0001, $\eta_p^2 = 0.57$; laser × conflict: F(2,32)=5.43, p=0.0093, $\eta_p^2 = 0.25$) (*Figure 4a–b*). As expected, in the absence of PRC inhibition, animals spent more time by the dispenser during no conflict trials compared to high conflict (p=0.002) and low conflict (p=0.03) trials.

When considering the time spent by the dispenser as a proportion of the total time in the goal box, PRC-inhibited rats spent a higher proportion of time at the dispenser during high conflict trials compared with low conflict (p=0.0008) and no conflict (p=0.0012) trials, as well as high conflict trials completed with the laser off (p=0.015; laser: F(1,18)=7.80, p=0.012, $\eta_p^2 = 0.30$, conflict: F(1.79,28.65)=11.19, p=0.00038, $\eta_p^2 = 0.41$, laser × conflict: F(2,32)=3.34, p=0.048, $\eta_p^2 = 0.17$) (*Figure 4c*). Laser off animals also spent proportionally more time by the dispenser during no conflict trials compared with low conflict trials (p=0.017), with no difference between no conflict and high conflict trials (p=0.89). To further investigate the impact of PRC inhibition on reward approach, we also analyzed the LTE into the object, neutral and goal boxes across the two levels of conflict that included an appetitive object (e.g., App+Av; App+Neu). Animals were significantly faster to enter all compartments under 'laser on' conditions (laser: F(1,16)=9.67, p=0.007, $\eta_p^2 = 0.38$), which was likely driven by the PRC inhibition group data (laser × construct: F(1,16)=3.92, p=0.065, $\eta_p^2 = 0.20$; laser × construct × box × conflict: F(2,32)=0.2, p=0.82; *Figure 2—figure supplement 1* a,c). However, all rats exhibited longer latencies to enter the boxes during the high conflict condition, compared with the low conflict test (conflict: F(1,16)=10.47, p<0.005, $\eta_p^2 = 0.40$). Object exploration was also comparable between high vs. low conflict conditions, with PRC inhibition having no impact on the durations of object exploration (conflict: F(1,16)=0.003, p=0.96, $\eta_p^2 = 0.0001$; laser: F(1,16)=0.034, p=0.86, $\eta_p^2 = 0.002$; laser × construct: F(1,16)=0.056, p=0.82, $\eta_p^2 = 0.003$).

Collectively, these findings indicate that the PRC inhibition-induced increase in approach behavior during the high and low conflict tests arises from alterations in conflict processing, rather than impairments in valence retrieval or novel stimulus recombination processing. Furthermore, in the absence of the normal PRC functioning, animals spent more time by the sucrose dispenser and traversed through the runway more quickly, reflecting a greater anticipation for a reward outcome.

To rule out encoding impairments leading to the observed PRC inhibition effects, a separate control cohort of animals (N = 16; ArchT = 8; GFP = 8) was trained on the runway paradigm, wherein the optogenetic inhibition was applied when the animal entered the neutral box (i.e., *after* object exploration) and prior to the choice period (*Figure 5a*).

All rats acquired the object-outcome associations successfully by test 2, with rats spending the most time in the goal box after exposure to the appetitive object pair (p<0.001), and the least time after aversive object pair exposure (p<0.001), compared to goal box time in neutral trials (valence: F(2,28)=231.3, p<0.001, $\eta_p^2 = 0.94$; *Figure 5b*); rats exhibited a significant increase in goal box time during appetitive trials and decrease in aversive trials between the two acquisition tests (valence × test: F(2,28)=54.91, p<0.001, $\eta_p^2 = 0.80$). The pattern of valence acquisition was comparable between ArchT and GFP animals (construct: F(1,14)=1.09, p=0.3, $\eta_p^2 = 0.07$; valence × construct: F(2, 28)=1.53, p=0.23, $\eta_p^2 = 0.10$; test × construct: F(1, 14)=1.64, p=0.22, $\eta_p^2 = 0.10$).

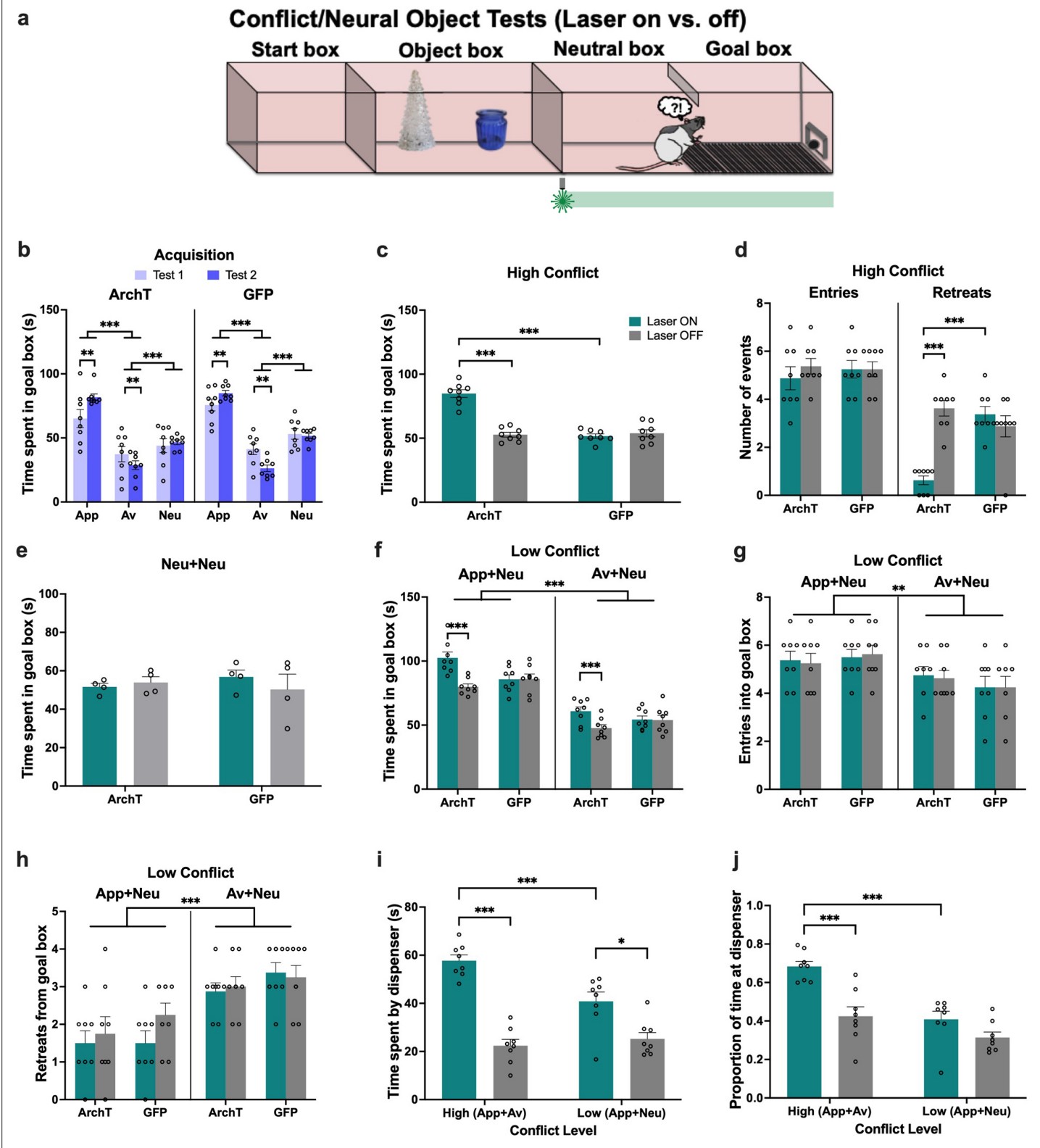

**Figure 5.** Perirhinal cortex (PRC) inhibition prior to the choice period of the runway task. In a separate cohort of rats (n=8 archaerhodopsin T [ArchT]; n=8 green fluorescent protein [GFP]), (**a**) optogenetic inhibition was applied upon confinement to the neutral box of the runway task, after object exploration but prior to goal box entry (i.e., choice behavior). (**b**) All animals demonstrated object-outcome associations by acquisition test 2. (**c–d**) PRC inhibition significantly increased time spent in the goal box and reduced the number of retreats in the high conflict test. (**e**) There was no effect of PRC

*Figure 5 continued on next page*

*Figure 5 continued*

inhibition on AA behavior in the neutral test. (**f**) Similar to the high conflict test, PRC inhibition increased time spent in the goal box in both low conflict tests. (**g–h**) PRC inhibition did not impact the number of entries or retreats in the App+Neu or Av+Neu low conflict tests, although there was a main effect of valence, with a greater number of entries for App+Neu and more retreats for Av+Neu. (**i–j**) PRC-inhibited rats spent more time by the sucrose dispenser, as measured by total time or proportion of time, in the high and low conflict tests but not the no conflict test. All figures show mean values ± SEM. Three-way ANOVA was conducted for all data shown shown except for data in panels c-e and i-j, which were subjected to two-way ANOVA. Significant interactions were followed up with post-hoc tests with Bonferroni correction. ***p<0.001, **p<0.01, *p<0.05.

The online version of this article includes the following figure supplement(s) for figure 5:

**Figure supplement 1.** Additional perirhinal cortex (PRC) rat data with optogenetic inhibition during the choice period for the approach-avoidance (AA) conflict runway task.

When animals were exposed to a high conflict object pair, optogenetic inhibition of PRC prior to goal box access significantly increased time spent in the goal box compared to trials without inhibition, and compared to GFP control animals (laser × construct: $F(1,14)=81.3$, $p<0.001$, $\eta_p^2=0.85$; all post hoc: $p<0.001$) (**Figure 5c**). Turning the laser on following neutral box entry did not impact time spent in the goal box when compared with animals that received whole-session laser application (laser timing: $F(1,34)=1.26$, $p=0.27$, $\eta_p^2=0.04$; laser timing × laser: $F(1,34)=0.06$, $p=0.81$, $\eta p2=0.002$; laser timing × construct: $F(1, 34)=0.12$, $p=0.7$, $\eta_p^2=0.005$), suggesting that PRC mediates object-based motivational conflict rather than encoding or retrieving representations of highly conflicting objects. The difference score of the two conflict test sessions (laser on–laser off) revealed that laser treatment increased goal box time for ArchT animals only (ArchT: 32.13±5.95, GFP: –2.22±8.98, $t(14) = 9.02$, $p<0.001$).

PRC inhibition during the choice period led to a significant decrease in the number of retreats from the goal box (laser × construct: $F(1,14) = 38.11$, $p<0.001$, $\eta_p^2 = 0.73$), while having no effect on the number of goal box entries (laser × construct: $F(1,14)=0.88$, $p=0.37$, $\eta_p^2 = 0.06$) (**Figure 5d**), the LTE each of the object, neutral, and goal boxes (laser × construct: $F(1,14)=0.17$, $p=0.69$, $\eta_p^2 = 0.01$; laser × construct × box: $F(2,28)=0.26$, $p=0.77$, $\eta_p^2 = 0.02$), or exploration of appetitive and aversive objects (laser × construct: $F(1,14) = 0.007$, $p=0.93$, $\eta_p^2 = 0.001$; laser × construct × object: $F(1,14)=0.05$, $p=82$, $\eta_p^2 = 0.004$) (**Figure 5—figure supplement 1a–b**). PRC inhibition had no effect on the time spent in the goal box during the neutral object test (laser × construct: $F(1,16)=0.87$, $p=0.37$, $\eta_p^2 = 0.07$; **Figure 5e**).

After a refresher conditioning session, rats underwent a low conflict recombination test. Similar to rats that received whole-session PRC inhibition, inhibiting PRC during the choice period increased the time spent in the goal box for both App-Neu and Av-Neu object pairs (laser × construct: $F(1,14)=15.77$, $p<0.005$, $\eta_p^2=0.53$) (**Figure 5f**). The laser treatment difference score analysis showed that ArchT animals spent more time in the goal box when faced with App-Neu (ArchT: 22.58±11.72, GFP: –0.34±9.22, $t(14) = 4.35$, $p=0.001$), but not Av-Neu (ArchT: 13.17±15.35, GFP: 0.40±16.71, $t(14) = 1.59$, $p=0.13$) pairings. PRC inhibition had no effect on the number of entries or retreats from the goal box during the choice period (entries: laser × construct: $F(1,14)=0.14$, $p=0.71$, $\eta_p^2=0.01$; retreats: laser × construct: $F(1,14)=0.24$, $p=0.64$, $\eta_p^2=0.05$) (**Figure 5g–h**). PRC inhibition also had no impact on the LTE for any of the runway boxes (laser × construct: $F(1,14)=1.91$, $p=0.19$, $\eta_p^2=0.10$; laser × construct × box: $F(2,28)=0.11$, $p=0.11$, $\eta_p^2=0.07$), or the exploration of the App-Neu and Av-Neu object pairings (laser × construct: $F(1,14)=1.42$, $p=0.25$, $\eta_p^2=0.09$; laser × construct × object: $F(1,14)=0.1$, $p=0.92$, $\eta_p^2=0.001$), with comparable object exploration between the valanced pairs (valence: $F(1,14)=2.36$, $p=0.15$, $\eta_p^2=0.14$; valence × construct: $F(1,14) = 0.001$, $p=0.97$, $\eta_p^2<0.0001$) (**Figure 5—figure supplement 1c–f**).

Notably, PRC inhibition during the choice period did not impair the ability to discriminate appetitive and aversive valences (valence: $F(1,14)=253.86$, $p<0.001$, $\eta_p^2 = 0.95$; laser × valence × construct: $F(1,14)=1.0$, $p=0.34$, $\eta_p^2 = 0.07$), with all animals spending significantly more time in the goal box for appetitive-neutral pairings than aversive-neutral ($p<0.001$, **Figure 5f**). Furthermore, all PRC rats made fewer entries ($p=0.001$) and exhibited more retreat behavior ($p<0.001$) when presented with an aversive-neutral pairing compared with an appetitive-neutral pairing (**Figure 5g–h**). Thus, valence recall was intact in PRC-inhibited rats.

Finally, PRC-inhibited ArchT animals spent significantly more time by the sucrose dispenser after encountering both high conflict ($p<0.001$) and low conflict ($p<0.01$) stimuli, compared to test

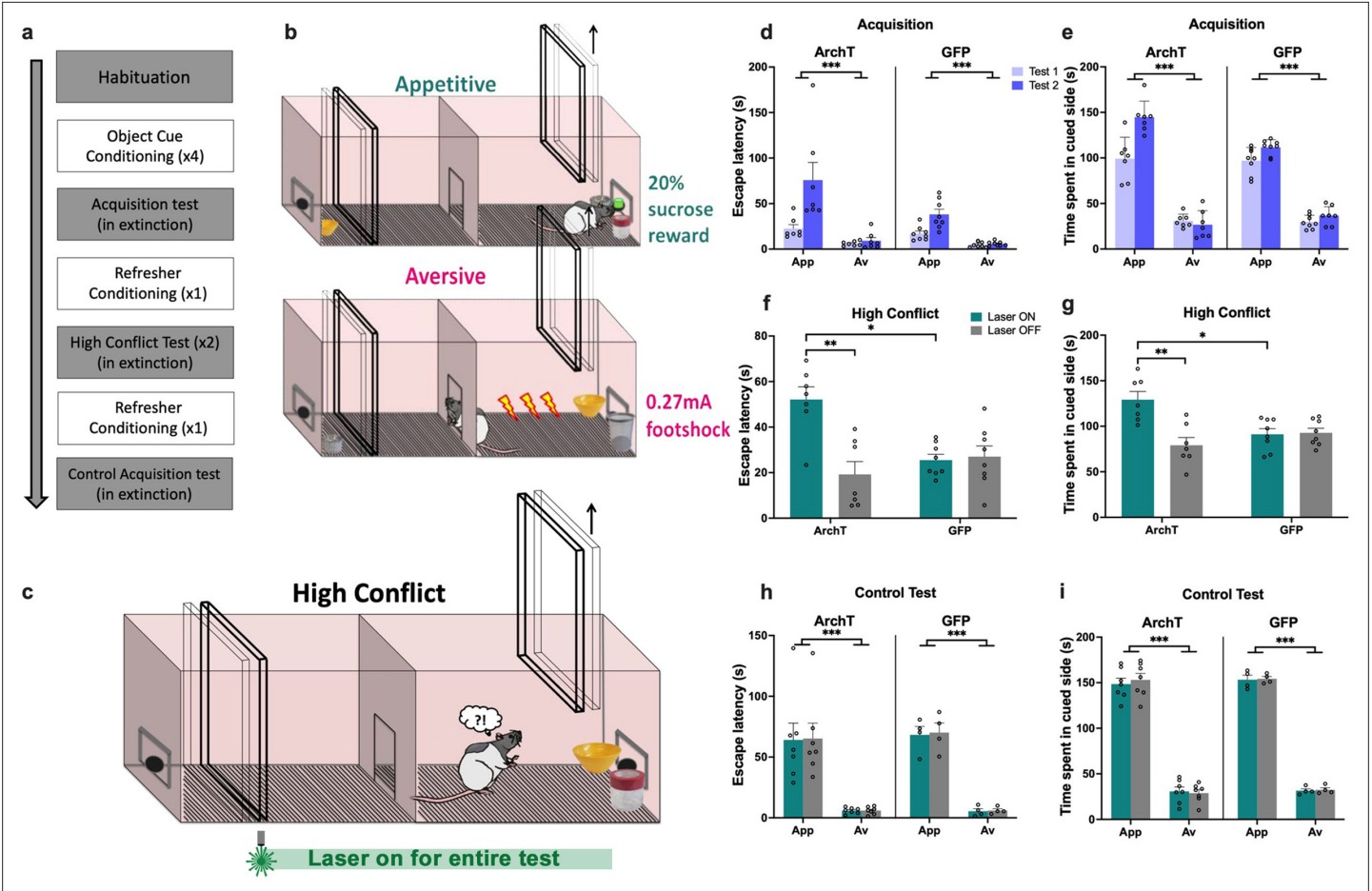

**Figure 6.** Impact of perirhinal cortex (PRC) inhibition on object approach-avoidance (AA) conflict shuttle box task performance. (**a**) Timeline of paradigm. (**b**) Rats (n=7 archaerhodopsin T [ArchT]; n=8 green fluorescent protein [GFP]) first learned to stay to receive a reward when exposed to an appetitive object pair, and to escape to avoid footshock when exposed to an aversive object pair. (**c**) Rats were then exposed to a high conflict object pairing in extinction. (**d–e**) All rats demonstrated intact acquisition of AA behavior. (**f–g**) PRC inhibition led to an increased escape latency in the high conflict test and a greater amount of time spent in the cued side of the shuttle box. (**h–i**) PRC inhibition did not impact escape latency or time spent in the cued side in a subsequent control no conflict test. All figures show mean values ± SEM. Three-way ANOVA was conducted for data shown in d-e, h-i and two-way ANOVA was conducted for data in f-g. Post-hoc tests with Bonferroni correction were conducted to further investigate significant interactions in all datasets. ***p<0.001, **p<0.01, *p<0.05.

sessions without optogenetic inhibition (laser: $F_{(1,7)}$=130.11, p<0.0001, $\eta_p^2$ = 0.95; laser × conflict: $F_{(1,7)}$=10.70, p=0.014, $\eta_p^2$ = 0.61) (*Figure 5i*). PRC-inhibited rats also spent a higher proportion of time by the dispenser during high conflict trials compared with low conflict (p<0.0001) trials, as well as high conflict trials completed with the laser off (p=0.001; laser: $F_{(1, 7)}$=34.10, p=0.001, $\eta_p^2$ = 0.83; conflict: $F_{(1,7)}$=24.41, p=0.002, $\eta_p^2$ = 0.78; laser × conflict: $F_{(1,7)}$=5.95, p=0.045, $\eta_p^2$ = 0.46) (*Figure 5j*). An analysis of the LTE data for each of the boxes between high and low conflict sessions revealed that PRC inhibition did not change LTE behavior (laser: $F_{(1,14)}$=0.803, p=0.39, $\eta_p^2$ = 0.05; laser × construct: $F_{(1,14)}$=1.48, p=0.24, $\eta_p^2$ = 0.1), with all rats exhibiting overall longer latencies to enter all boxes during the high conflict test compared with the low conflict test (p=0.043; conflict: $F_{(1,14)}$=9.99, p=0.007, $\eta_p^2$ = 0.42; construct: $F_{(1,14)}$=0.17, p=0.69, $\eta_p^2$ = 0.01; conflict × construct: $F_{(1,14)}$=0.28, p=0.60, $\eta_p^2$ = 0.02; laser × construct × box × conflict: $F_{(2,28)}$=0.16, p=0.85).

## Optogenetic inhibition of PRC decreases avoidance behavior to motivational conflict represented by objects in a shuttle box paradigm

To investigate whether the observed increase in goal box preference time exhibited by PRC-inhibited rats was specific to the conditions elicited by the runway task, the same PRC rats completed another

novel behavioral paradigm conducted in a modified two-way active avoidance 'shuttle box' apparatus (*Figure 6a–c*). In the first phase of the paradigm, appetitive or aversive object cue pairs used in the 'no conflict' recombination test in the runway task were placed in opposite ends of a two-compartment apparatus, behind transparent barriers. In each trial, animals were allowed to visually sample the objects for 1 min, after which the transparent barrier was raised to allow animals access to the associated outcome (sucrose or shock). At the same time, a central guillotine door dividing the two compartments was raised to give rats the opportunity to shuttle into the opposite compartment to escape the shock outcome associated with the aversive object cue pair. Acquisition of escape behavior was assessed after four (test 1) and eight (test 2) conditioning sessions, without laser treatment. PRC rats demonstrated the expected behavior by test 2, with rats exhibiting significantly shorter escape latencies after exposure to the aversive object pair compared with exposure to the appetitive object pair (valence: $F(1,13)=45.97$, $p<0.0001$, $\eta_p^2 = 0.78$; valence × test: $F(1,13)=16.17$, $p=0.001$, $\eta_p^2 = 0.55$) (*Figure 6d*). The pattern of escape acquisition was comparable between ArchT and GFP animals (construct: $F(1,13)=4.55$, $p=0.052$, $\eta_p^2 = 0.26$; valence × construct: $F(1,13)=4.22$, $p=0.06$, $\eta_p^2 = 0.25$; test × construct: $F(1,13)=2.69$, $p=0.13$, $\eta_p^2 = 0.17$). PRC rats also spent more time in the outcome-associated area (stay behavior) when exposed to the appetitive object pair compared with the aversive object pair (valence: $F(1,13)=456.04$, $p<0.0001$, $\eta_p^2 = 0.97$; valence × test: $F(1,13)=21.18$, $p<0.001$, $\eta_p^2 = 0.62$) (*Figure 6e*). This pattern of stay behavior was also comparable between ArchT and GFP animals (construct: $F(1,13)=3.57$, $p=0.082$, $\eta_p^2 = 0.22$; valence × construct: $F(1,13)=8.17$, $p=0.013$, $\eta_p^2 = 0.39$; test × construct: $F(1,13)=1.65$, $p=0.22$, $\eta_p^2 = 0.11$).

When animals were next exposed to a high conflict pairing, optogenetic inhibition of PRC (laser on) significantly prolonged escape latencies compared with trials completed without inhibition (laser off; $p<0.001$), and compared to GFP control animals ($p<0.005$) with laser on and off (laser × construct: $F(1,13)=18.97$, $p=0.001$, $\eta_p^2 = 0.59$) (*Figure 6f*), indicative of a decrease in avoidance behavior under motivational conflict. Furthermore, despite making a comparable number of re-entries into the outcome-associated area during the test session (laser × construct: $F(1,13)=0.99$, $p=0.34$, $\eta_p^2 = 0.07$), PRC-inhibited animals spent significantly longer in the outcome-associated area containing the conflict object pair compared with trials completed without inhibition (laser off; $p<0.001$), and compared to GFP control animals ($p<0.005$) with laser on and off (laser × construct: $F(1,13)=12.32$, $p=0.004$, $\eta_p^2 = 0.49$) (*Figure 6g*). An analysis of the difference score of the two conflict test sessions (laser on–laser off) revealed that laser treatment both decreased escape latency (ArchT: $32.91\pm19.85$, GFP: $-1.54\pm9.8$, Mann-Whitney U=2, ArchT = 7, GFP = 8, $p=0.001$, two-tailed), and increased cumulative duration in the outcome-associated area (ArchT: $50.04\pm36.64$, GFP: $-1.57\pm18.70$, Mann-Whitney U=7, $p=0.014$, two-tailed) for ArchT animals compared with GFP.

Following 'refresher conditioning,' PRC rats completed a set of control tests, in which conditioned cue acquisition tests with and without PRC inhibition were completed. PRC inhibition had no effect on the duration of escape latencies, defined as the length of the first 'stay' only in the cued side, for either appetitive or aversive valences (laser: $F(1,13)=0.88$, $p=0.37$, $\eta_p^2 = 0.06$; laser × construct: $F(1,13)=0.15$, $p=0.71$, $\eta_p^2 = 0.01$; valence × construct: $F(1,13)=0.04$, $p=0.84$, $\eta_p^2 = 0.003$), with all rats exhibiting significantly shorter escape latencies for aversive than appetitive trials (valence: $F(1,13)=84.84$, $p<0.001$, $\eta_p^2 = 0.87$; construct: $F(1,13)=0.06$, $p=0.82$, $\eta_p^2 = 0.004$) (*Figure 6h*). Similarly, PRC inhibition had no effect on the amount of time spent in the cued side of the apparatus, defined as the total time spent across multiple stays in the chamber (laser: $F(1,13)=0.59$, $p=0.46$, $\eta_p^2 = 0.04$; laser × construct: $F(1,13)=0.003$, $p=0.96$, $\eta_p^2 < 0.001$; valence × construct: $F(1,13)=0.22$, $p=0.65$, $\eta_p^2 = 0.02$), with all rats spending significantly less time in the cued side during aversive than appetitive trials (valence: $F(1,13)=470.47$, $p<0.0001$, $\eta_p^2 = 0.97$; construct: $F(1,13)=0.55$, $p=0.47$, $\eta_p^2 = 0.04$) (*Figure 6i*). Collectively, these results suggest that PRC inhibition led to both a decrease in avoidance behavior and/or an increase in approach/stay behavior under motivational conflict, and that PRC-inhibited animals could readily discriminate between appetitive and aversive object cues.

As with the runway task, a separate cohort of animals received optogenetic inhibition immediately after object exploration (i.e., prior to the raising of the central guillotine door; *Figure 7*). All rats exhibited significantly shorter escape latencies after exposure to the aversive object pair compared with exposure to the appetitive object pair by the second acquisition test (valence: $F(1,14)=123.22$, $p<0.001$, $\eta_p^2 = 0.9$; valence × test: $F(1,14)=116.05$, $p<0.001$, $\eta_p^2 = 0.89$) (*Figure 7b*). The pattern of escape acquisition was comparable between ArchT and GFP animals (construct: $F(1,14)=0.23$, $p=0.63$,

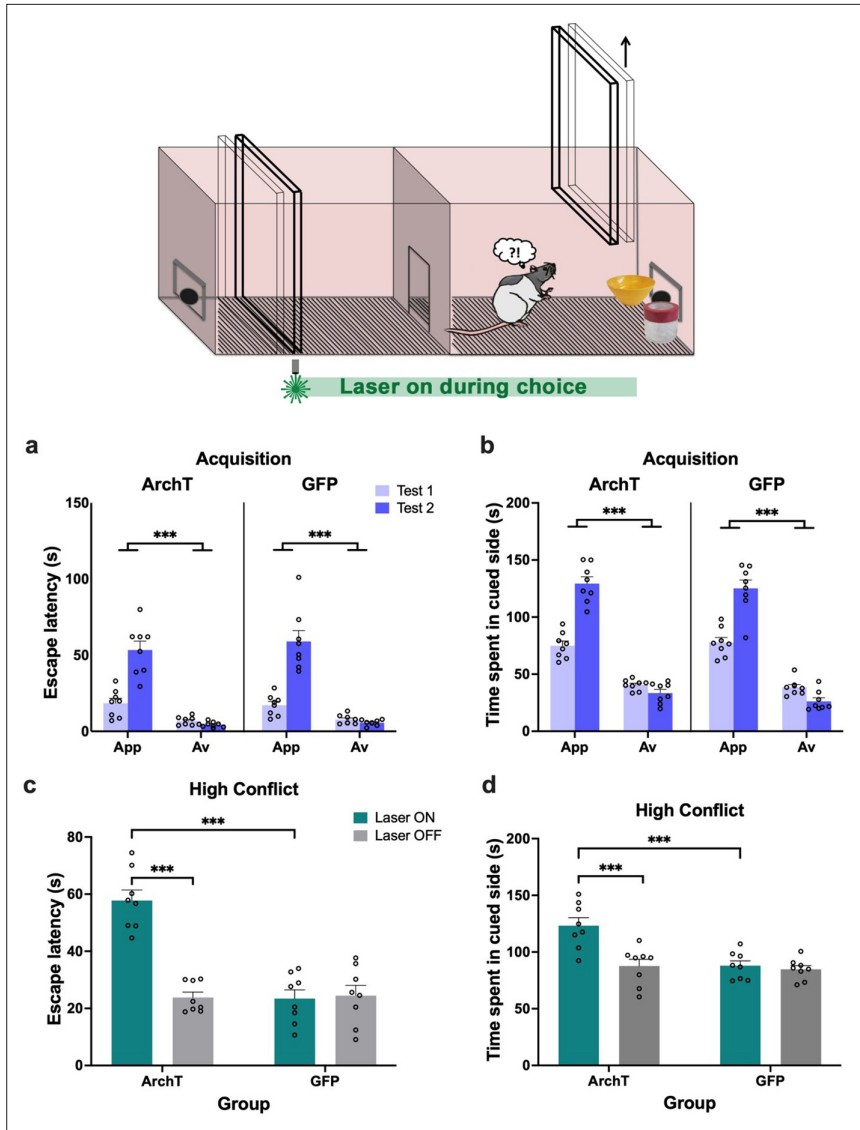

**Figure 7.** Perirhinal cortex (PRC) inhibition during the choice period of the object approach-avoidance (AA) conflict shuttle box task. (**a–b**) All rats (n=8 archaerhodopsin T [ArchT]; n=8 green fluorescent protein [GFP]) demonstrated intact acquisition of AA behavior. (**c–d**) PRC inhibition led to an increased escape latency in the high conflict test and a greater amount of time spent in the cued side of the shuttle box. All figures show mean values ± SEM. Three-way ANOVA was conducted for data shown in a-b, and two-way ANOVA was conducted for data in c-d. Post-hoc tests with Bonferroni correction were conducted to further investigate significant interactions. ***p<0.001.

$\eta_p^2 = 0.02$; valence × construct: F(1,14)=0.04, p=0.85, $\eta_p^2 = 0.003$; test × construct: F(1,14)=1.06, p=0.32, $\eta_p^2 = 0.07$). Rats also spent more time in the outcome-associated area (stay behavior) when exposed to the appetitive object pair compared with the aversive object pair (valence: F(1,14)=419.32, p<0.0001, $\eta_p^2 = 0.97$; valence × test: F(1,14)=140.61, p<0.0001, $\eta_p^2 = 0.91$) (**Figure 7b**). This pattern of stay behavior was also comparable between ArchT and GFP animals (construct: F(1,9)=2.66, p=0.13, $\eta_p^2 = 0.23$; valence × construct: F(1,14)=0.35, p=0.56, $\eta_p^2 = 0.02$; test × construct: F(1,14)=0.95, p=0.35, $\eta_p^2 = 0.06$).

When animals were exposed to a high conflict object pair, optogenetic inhibition of PRC (laser on) during the choice period significantly prolonged escape latencies compared with trials completed without inhibition (laser off; p<0.001), and compared to GFP control animals (p<0.001) with laser on and off (laser × construct: F(1,14)=47.74, p<0.001, $\eta_p^2 = 0.77$) (**Figure 7c**), indicative of a decrease in avoidance behavior under motivational conflict. Furthermore, despite making a comparable number

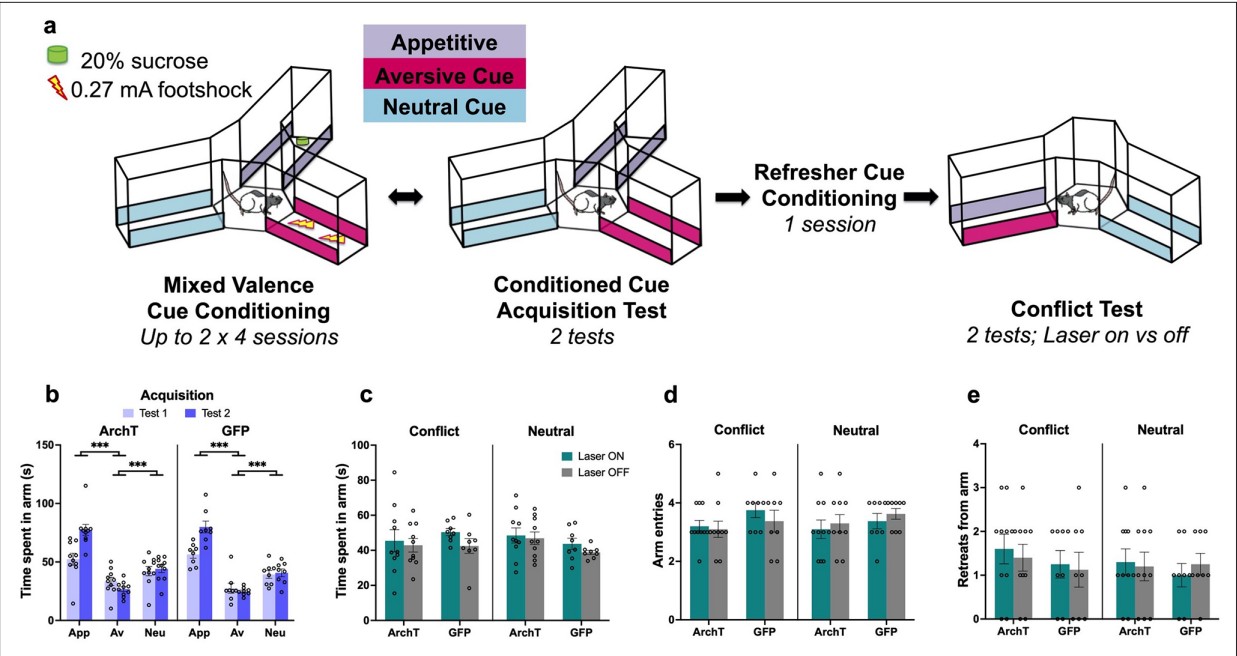

**Figure 8.** No effect of perirhinal cortex (PRC) inhibition on contextual approach-avoidance (AA) conflict behavior. (**a**) PRC rats (n=10 archaerhodopsin T [ArchT]; n=8 green fluorescent protein [GFP]) underwent a contextual AA task known to be ventral hippocampus (vHPC)-dependent, in which they first learned the outcomes associated with appetitive, aversive, and neural cues and then underwent a conflict test in extinction. (**b**) Both ArchT and GFP rats demonstrated successful valence acquisition. (**c–e**) PRC inhibition had no effect on choice behavior during the conflict test. All figures show mean values ± SEM. Three-way ANOVA was conducted for all data shown, followed by post-hoc tests with Bonferroni correction in panel b data. ***p<0.001.

of re-entries into the outcome-associated area during the test session (laser × construct: F(1,14)=2.07, p=0.17, $\eta_p^2$ = 0.13), PRC-inhibited animals spent significantly longer in the outcome-associated area containing the conflict object pair compared with trials completed without inhibition (laser off; p<0.001), and compared to GFP control animals (p<0.005) with laser on and off (laser × construct: F(1,14)=14.98, p=0.002, $\eta_p^2$ = 0.52) (*Figure 7d*). An analysis of the difference score of the two conflict test sessions (laser on–laser off) revealed that laser treatment both decreased escape latency (ArchT: 33.93±10.09, GFP: −1.02±10.14, t(14) = 6.91, p<0.001), and increased cumulative duration in the outcome-associated area (ArchT: 35.55±19.35, GFP: 3.21±13.58, t(14) = 6.91, p<0.001 t(14)=3.87, p=0.002) for ArchT animals compared with GFP. These findings align with the data obtained from animals receiving continuous PRC inhibition throughout the shuttle box test, indicating that impairments in encoding or valence recall when faced with high motivational conflict are unlikely to contribute to the decreased avoidance behavior following PRC inhibition.

## PRC inhibition does not impact contextual AA conflict processing

To investigate whether the observed alteration in object-based AA conflict processing extended to conflict represented by 'context-like' cues, we administered an established Y-maze AA task that is vHPC-dependent, with large vHPC lesions and ventral CA3 (vCA3) or dentate gyrus inactivation increasing approach behavior in the face of high AA conflict (*Schumacher et al., 2018*; *Schumacher et al., 2016Yeates et al., 2020*).

Rats first learned the valences of three visuotactile cues (appetitive, aversive, and neutral) that spanned the length of three different maze arms (*Figure 8a*). Analysis of time spent in a given arm after four (test 1) and eight (test 2) conditioning sessions, without laser treatment, revealed that all rats acquired the cue-outcome associations successfully by test 2 (valence: F(2,32)=146.02, p<0.0001, $\eta_p^2$ = 0.9; valence × test: F(2,32)=24.26, p<0.0001, $\eta_p^2$ = 0.6), regardless of construct (construct: F(1,16)=0.17, p=0.69, $\eta_p^2$ = 0.01; test × construct: F(1,16) = 0.01, p=0.92, $\eta_p^2$ = 0.001; valence × construct: F(2,32)=1.51, p=0.24, $\eta_p^2$ = 0.09) by spending significantly more time in the appetitive arm (both tests p<0.001) and less time in the aversive arm (both p≤0.003) compared with the neutral arm (*Figure 8b*). Rats then completed two AA conflict tests during which they could freely explore between

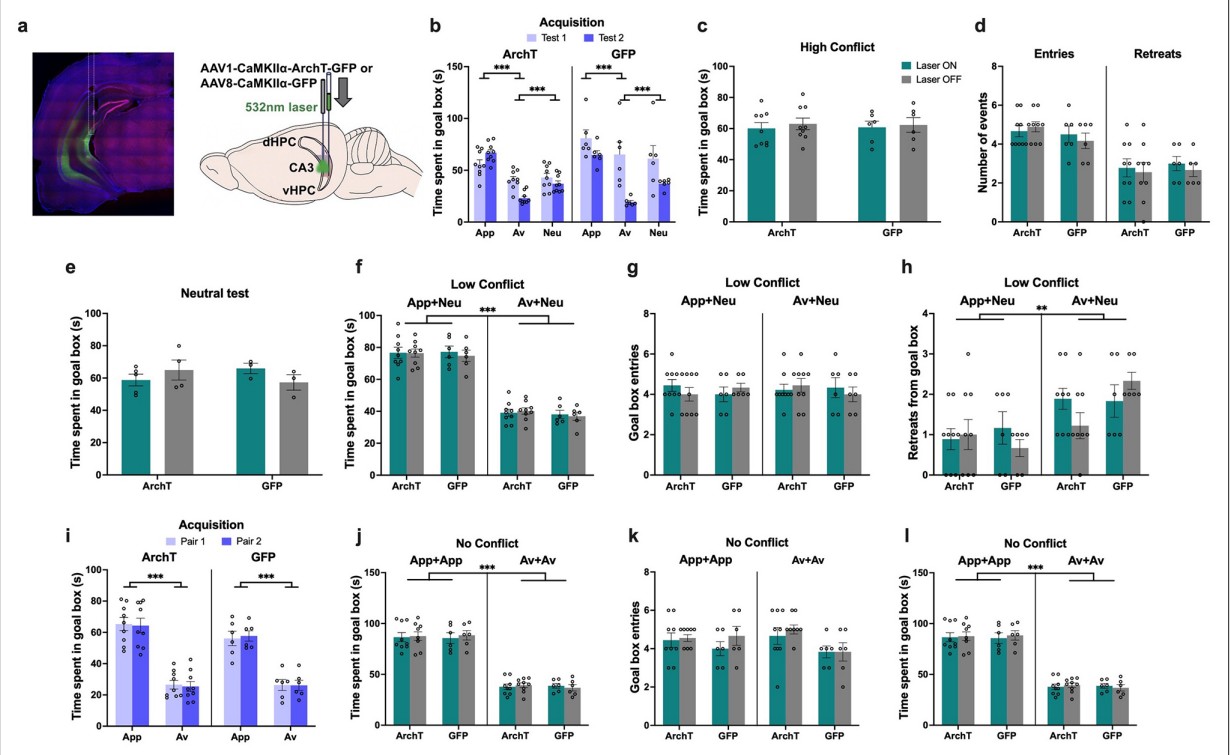

**Figure 9.** No effect of ventral CA3 (vCA3) inhibition on object approach-avoidance (AA) conflict runway task performance. (**a**) Rats injected with archaerhodopsin T (ArchT) (n=9) or green fluorescent protein (GFP) (n=6) in the vCA3 underwent the object runway task. (**b**) Both groups learned the appetitive (App), aversive (Av), and neutral (Neu) object pairs successfully. (**c–h**) vCA3 inhibition did not impact any behavioral measure on the high conflict, neutral, or low conflict recombination tests. (**i**) Rats successfully learned a new set of object pairs for the no conflict recombination tests. (**j–l**) vCA3 inhibition did not impact performance on the no conflict recombination tests. All figures show mean values ± SEM. Three-way ANOVA and post-hoc tests with Bonferroni correction (where significant interactions were found) were conducted for all data shown shown except for data in panels c–e, which were subjected to two-way ANOVA. ***p<0.001, **p<0.01.

a conflict arm containing a combination of appetitive and aversive cues, and a neutral arm containing the neutral cue. PRC inhibition did not significantly change the time spent in either arm (laser: $F_{(1,16)}$ = 4.06, p=0.060, $\eta_p^2$ = 0.2; laser × construct: $F_{(1,16)}$=1.06, p=0.32, $\eta_p^2$ = 0.06) (*Figure 8c*), and both ArchT and GFP animals spent a comparable duration of time in both the conflict and neutral arms (arm: $F_{(1,16)}$=0.09, p=0.77, $\eta_p^2$ = 0.005; construct: $F_{(1,16)}$=0.31, p=0.58, $\eta_p^2$ = 0.02; arm × construct: $F_{(1,16)}$=2.47, p=0.14, $\eta_p^2$ = 0.13). Moreover, PRC inhibition did not significantly alter the number of entries made into or retreats from either the conflict or neutral arms (entries: laser: $F_{(1,16)}$=0.001, p=0.98, $\eta_p^2$ <0.001; construct: $F_{(1,16)}$=3.36, p=0.09, $\eta_p^2$ = 0.17; laser × construct: $F_{(1,16)}$=0.07, p=0.8, $\eta_p^2$ = 0.004; *retreats*: laser: $F_{(1,16)}$=0.03, p=0.86, $\eta_p^2$ = 0.002; construct: $F_{(1,16)}$=0.5, p=0.49, $\eta_p^2$ = 0.3; laser × construct: $F_{(1,16)}$=0.2, p=0.66, $\eta_p^2$ = 0.12) (*Figure 8d–e*). The number of entries and retreats also did not differ between conflict and neutral arms (*entries*: arm: $F_{(1,16)}$=0.001, p=0.98, $\eta_p^2$ <0.001; *retreats*: arm: $F_{(1,16)}$=1.19, p=0.29, $\eta_p^2$ = 0.07) for both ArchT and GFP animals (*entries*: arm × construct $F_{(1,16)}$=0.07, p=0.8, $\eta_p^2$ = 0.004; *retreats*: arm × construct: $F_{(1,16)}$=0.43, p=0.52, $\eta_p^2$ = 0.03). Thus, while PRC is critical for AA conflict processing associated with discrete objects, it may play a minimal role in contextual AA conflict.

## Optogenetic inhibition of vCA3 does not affect object-associated AA conflict processing

To investigate whether the vHPC plays a role in object-associated AA conflict processing, rats with either ArchT or GFP and optical fiber implants in the vCA3 (*Figure 9a*) were administered the object runway paradigm. vCA3 rats demonstrated successful valence learning across the two acquisition tests (valence: $F_{(2,26)}$=160.44, p<0.0001, $\eta_p^2$ = 0.93; valence × test: $F_{(2,26)}$=23.73, p<0.0001, $\eta_p^2$ = 0.65) (*Figure 9b*), and at test 2 spent the most time in the goal box after appetitive object exposure

and the least time after aversive object exposure in comparison to neutral trials (both p<0.001). Object valence acquisition was similar for ArchT and GFP animals (construct: F(1,13)=3.85, p=0.07, $\eta_p^2$ = 0.23; valence × construct: F(2,26)=0.20, p=0.82, $\eta_p^2$ = 0.02) with the exception that ArchT rats spent less time in the goal box compared to GFP rats during acquisition test 1 (test × construct: F(1,13)=7.13, p=0.019, $\eta_p^2$ = 0.35; test 1 construct: F(1,13)=10.94, p=0.006, $\eta_p^2$ = 0.46). ArchT and GFP animals did not, however, differ at the end of training (test 2 construct: F(1,13)=2.21, p=0.16, $\eta_p^2$ = 0.15).

In contrast to PRC inhibition, there was no effect of vCA3 inhibition on time spent in the goal box in the high conflict test session (laser: F(1,13)=2.92, p=0.11, $\eta_p^2$ = 0.18; construct: F(1,13)=0.001, p=0.99, $\eta_p^2$ <0.001; laser × construct: F(1,13)=0.93, p=0.60, $\eta_p^2$ = 0.02) (*Figure 9c*). Indeed, an overall comparison across regions revealed that the impact of laser manipulation was significantly different between PRC and vCA3 animals (laser × region × construct: F(1,29)=22.83, p<0.0001, $\eta_p^2$ = 0.44). vCA3 inhibition also did not impact other behavioral measures (*Figure 9d*) including the number of retreats (laser × construct: F(1,13)=0.03, p=0.86, $\eta_p^2$ = 0.003), entries made into the goal box (laser × construct: F(1,13)=0.58, p=0.46, $\eta_p^2$ = 0.04), and LTE (laser × construct: F(1,13)=0.37, p=0.55, $\eta_p^2$ = 0.03). vCA3 inhibition also did not affect time spent in the goal box when rodents were presented with the neutral object pair (laser: F(1,11)=0.03, p=0.90, $\eta_p^2$ = 0.03; laser × construct: F(1,11)=2.36, p=0.15, $\eta_p^2$ = 0.18) (*Figure 9e*).

Similar to the high conflict test, vCA3 inhibition did not impact the time spent in the goal box during the low conflict test (laser: F(1,13)=0.11, p=0.75, $\eta_p^2$ = 0.008; laser × construct: F(1,13)=0.29, p=0.60, $\eta_p^2$ = 0.02) (*Figure 9f*) and crucially, there was a significant difference between vCA3 and PRC rats when compared directly (laser × area × construct: F(1,25)=5.13, p=0.032, $\eta_p^2$ = 0.17). All vCA3 animals could readily discriminate between valences as reflected by time spent in the goal box (valence: F(1,13)=550.18, p<0.0001, $\eta_p^2$ = 0.98; valence × construct: F(1,13)=0.25, p=0.63, $\eta_p^2$ = 0.02) and retreats (valence: F(1,13)=13.80, p=0.003, $\eta_p^2$ = 0.52). There was, however, no effect of vCA3 inhibition on entries into and retreats from the goal box (both valence × construct: F(1,13)≤0.55, p≥0.47, $\eta_p^2$≤ 0.09) (*Figure 9g–h*).

Following successful acquisition of a new set of appetitive and aversive object pairs (valence: F(1,13)=361.37, p<0.0001, $\eta_p^2$ = 0.97; pairing: F(1,13)=0.006, p=0.94, $\eta_p^2$ <0.001; valence × pairing: F(1,13)=0.19, p=0.67, $\eta_p^2$ = 0.02; pairing × construct: F(1, 13)=0.14, p=0.72, $\eta_p^2$ = 0.01) (*Figure 9i*), vCA3 inhibition also did not impact time spent in the goal box in the within-valence no conflict recombination test (laser: F(1,13)=0.14, p=0.72, $\eta p^2$=0.01; laser × construct: F(1,13)=0.040, p=0.85, $\eta_p^2$ = 0.003) (*Figure 9j*), with entries into and retreats from the goal box also unaffected (both valence × construct: F(1,13)≤1.10, p≥0.31, $\eta_p^2$≤ 0.2) (*Figure 9k–l*). Both ArchT and GFP animals could discriminate between the two valences by spending more time in the goal box and retreating less for appetitive pairs compared to aversive pairs (valence: F(1,13)=481.57, p<0.0001, $\eta_p^2$ = 0.97; valence × construct: F(1,13)=0.03, p=0.88, $\eta_p^2$ = 0.002). In sum, vCA3 inhibition had no impact on any aspect of performance on the object AA runway task.

vCA3 rats also completed the object-based shuttle box, and all rats demonstrated successful establishment of an active avoidance response toward aversive, but not appetitive object pairings, across two acquisition tests, exhibiting significantly shorter escape latencies after exposure to the aversive object pair compared with exposure to the appetitive object pair (valence: F(1,13)=119.71, p<0.0001, $\eta_p^2$ = 0.90; valence × test: F(1,13)=92.12, p<0.0001, $\eta_p^2$ = 0.88; construct: F(1,13)=1.43, p=0.25, $\eta_p^2$ = 0.10; valence × construct: F(1,13)=1.18, p=0.30, $\eta_p^2$ = 0.08; test × construct: F(1,13)=0.03, p=0.87, $\eta_p^2$ = 0.002) (*Figure 10a*). All vCA3 rats also spent more in the outcome-associated area when exposed to the appetitive object pair compared with the aversive object pair (valence: F(1,13)=384.84, p<0.0001, $\eta_p^2$ = 0.97; valence × test: F(1,13)=9.41, p=0.0090, $\eta_p^2$ = 0.42; construct: F(1,13)=0.01, p=0.98, $\eta_p^2$ < 0.001; valence × construct: F(1,13)=0.68, p=0.42, $\eta_p^2$ = 0.05; test × construct: F(1,13)=1.28, p=0.28, $\eta_p^2$ = 0.09) (*Figure 10b*).

In contrast to PRC rats, there was no effect of vCA3 inhibition on escape latency behavior when exposed to the high conflict object pairing (laser: F(1,13)=0.48, p=0.5, $\eta_p^2$ = 0.04; construct: F(1,13)=0.03, p=0.87, $\eta_p^2$ = 0.002; laser × construct: F(1,13)=0.29, p=0.87, $\eta_p^2$ = 0.02) (*Figure 10c*). An overall comparison across regions revealed that the impact of laser manipulation was significantly different between PRC and vCA3 animals (laser × region × construct: F(1,22)=13.78, p=0.0012, $\eta_p^2$ = 0.39). vCA3 inhibition also had no effect on the time spent in the outcome-associated area containing

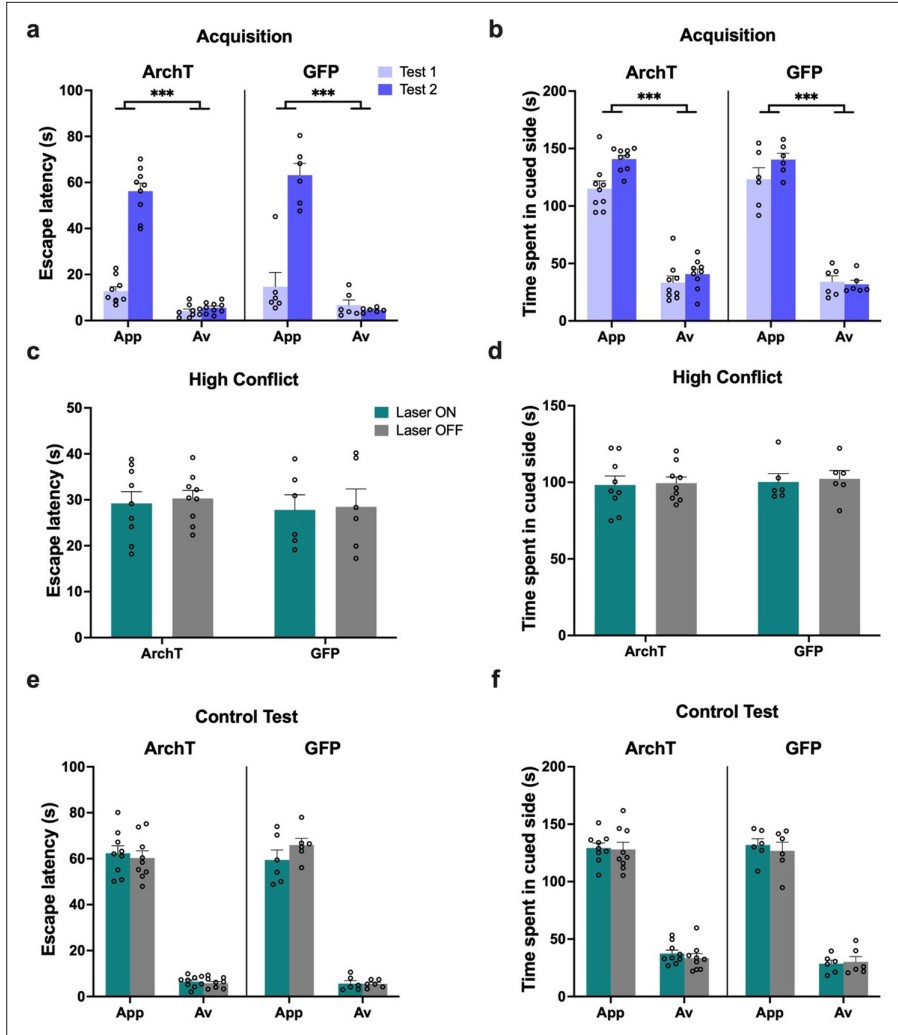

**Figure 10.** No impact of ventral hippocampus (vHPC) inhibition on object approach-avoidance (AA) conflict shuttle box task performance. (**a–b**) Both archaerhodopsin T (ArchT) (n=9) and green fluorescent protein (GFP) ventral CA3 (vCA3) (n=6) rats demonstrated intact acquisition of AA behavior on the shuttle box task. (**c–d**) vCA3 inhibition had no effect on escape latency or time spent in the cued side in the high conflict test. (**e–f**) vCA3 inhibition also did not impact behavior on the control no conflict test. All figures show mean values ± SEM. Three-way ANOVA, followed by post-hoc tests with Bonferroni correction was conducted for data shown in a-b and e-f, ***p<0.001, **p<0.01, and two-way ANOVA was conducted for data shown in c-d.

the conflict object pair (laser: $F_{(1,13)}$=0.26, p=0.62, $\eta_p^2$ = 0.02; construct: $F_{(1,13)}$=0.12, p=0.74, $\eta_p^2$ = 0.009; laser × construct: $F_{(1,13)}$=0.020, p=0.89, $\eta_p^2$ = 0.002) (**Figure 10d**). Similarly, there was a significant difference between vCA3 and PRC animals when compared directly (laser × region × construct: $F_{(1,22)}$=7.98, p=0.010, $\eta_p^2$ = 0.27).

vCA3 inhibition also had no effect on the duration of escape latencies for either appetitive or aversive valences during the control acquisition tests (laser: $F_{(1,13)}$=0.51, p=0.49, $\eta_p^2$ = 0.04; laser × construct: $F_{(1,13)}$=3.13, p=0.1, $\eta_p^2$ = 0.19; valence × construct: $F_{(1,13)}$=0.25, p=0.63, $\eta_p^2$ = 0.02), with all rats exhibiting significantly shorter escape latencies for aversive than appetitive trials (valence: $F_{(1,13)}$=826.68, p<0.0001, $\eta_p^2$ = 0.99; construct: $F_{(1,13)}$=0.03, p=0.88, $\eta_p^2$ = 0.002) (**Figure 10e**). Similarly, vCA3 inhibition had no effect on the amount of time spent in the cued side of the apparatus (laser: $F_{(1,13)}$=0.86, p=0.37, $\eta_p^2$ = 0.06; laser × construct: $F_{(1,13)}$=0.04, p=0.84, $\eta_p^2$ = 0.003; valence × construct: $F_{(1,13)}$=0.69, p=0.42, $\eta_p^2$ = 0.05), with all rats spending significantly less time in the cued side during aversive than appetitive trials (valence: $F_{(1,13)}$=565.77, p<0.0001, $\eta_p^2$ = 0.98; construct: $F_{(1,13)}$=0.30, p=0.59, $\eta_p^2$ = 0.02) (**Figure 10f**).

In sum, these results indicate that vCA3 inhibition had no impact on object-associated motivational behavior whether under AA conflict or the presence of appetitive or aversive cues alone.

## Optogenetic inhibition of the PRC and vCA3 reduces cFos+ expression

To confirm optogenetic inhibition of cellular activity in the PRC and vCA3, cFos immunohistochemistry was performed following a PRC-dependent task: novel object recognition (NOR), in which rats explored an open maze containing a novel and a familiar object; or a vHPC-dependent elevated plus maze (EPM), in which rats explored two anxiogenic open arms and two 'safe' closed arms.

For the NOR task, an analysis of the discrimination ratio (difference between novel and familiar object exploration divided by total exploration) revealed that laser-treated PRC ArchT animals exhibited a significant NOR impairment (laser: $F_{(1,34)}=45.6$, $p<0.0001$, $\eta_p^2 = 0.57$, construct: $F_{(1,34)}=28.46$, $p<0.0001$, $\eta_p^2 = 0.46$; laser × construct: $F_{(1,34)}=39.63$, $p<0.0001$, $\eta_p^2 = 0.54$) (*Figure 11a*) compared with laser-off ArchT and PRC-GFP control animals (all $p\leq0.001$). In the EPM, laser-treated vCA3-ArchT animals spent a significantly increased proportion of time in the open arms (laser: $F_{(1,11)}=8.02$, $p=0.003$, $\eta_p^2 = 0.42$; construct: $F_{(1,11)}=14.42$, $p=0.003$, $\eta_p^2 = 0.57$; laser × construct: $F_{(1,11)} = 7.1$, $p=0.022$, $\eta_p^2 = 0.39$) compared with laser-off Arch T and vCA3-GFP control animals (all $p\leq0.001$) (*Figure 11b*).

ArchT animals that completed either the NOR (PRC) or EPM (vCA3) tasks with the laser on prior to sacrifice demonstrated a significant reduction in cFos labeling in the PRC (laser: $F_{(1,34)}=98.11$, $p<0.0001$, $\eta_p^2 = 0.74$; construct: $F_{(1,34)}=56.49$, $p<0.0001$, $\eta_p^2 = 0.62$; laser × construct: $F_{(1,34)}=64.82$, $p<0.0001$, $\eta_p^2 = 0.66$) or vCA3 (laser: $F_{(1,11)}=5.81$, $p=0.035$, $\eta_p^2 = 0.35$; construct: $F_{(1,11)}=3.77$, $p=0.07$, $\eta_p^2 = 0.26$; laser × construct: $F_{(1,11)}=12.78$, $p=0.004$; $\eta_p^2 = 0.54$), compared with ArchT animals that completed the tasks with laser off and GFP control animals (all $p\leq0.001$) (*Figure 11c–e*). Thus, laser treatment in ArchT animals significantly reduced neural activation.

Finally, histological analysis confirmed robust bilateral expression of GFP/ArchT and optic fiber tip placement immediately dorsal to the viral injection site in all PRC animals (*Figure 11*). Thus, no exclusions were made based on optic fiber placement/viral expression. In the vCA3 group, data from three ArchT-expressing animals were excluded based on the viral expression and optic fiber placement presenting too medially to the CA3 subfield.

## Discussion

Using a set of original object-based AA paradigms, we found that optogenetic inhibition of the rodent PRC, but not the vCA3, resulted in a robust increase in approach bias during motivational conflict elicited by the presentation of discrete object pairs associated with non-matching affective values. In contrast, PRC inhibition did not disrupt behavior during a contextual vHPC-dependent AA task. Critically, PRC inhibition did not disrupt AA behavior in response to neutral stimuli or novel re-configurations of objects with the same valence. Furthermore, selective optogenetic inhibition of the PRC continuously throughout the entire test session or prior to the first goal box choice (and therefore post-object exploration) induced the same approach bias in choice behavior, suggesting that the observed impact of PRC inhibition on object-associated AA conflict processing was unlikely to be driven by impairments in mnemonic functioning.

Our finding that optogenetic inhibition of PRC, but not vCA3, disrupted object-associated motivational conflict behavior contrasts with a plethora of studies detailing a role for the rodent vHPC in the resolution of AA conflict (*Ito and Lee, 2016*). However, a common characteristic of these prior studies is the employment of spatial/contextual stimuli during testing. For example, when spatial locations in ethological tests of anxiety or contextual cues with learned incentive values in a Y-maze task have been used, animals with broad vHPC damage or subfield-specific inactivation of the vCA3 and DG exhibit increased approach behavior under motivational conflict, whereas vCA1-inhibited animals exhibit greater avoidance behavior (*Bannerman et al., 2003*; *Bannerman et al., 2002*; *Schumacher et al., 2018*; *Schumacher et al., 2016*; *Yeates et al., 2020*). The present study advances this work by utilizing discrete objects as target stimuli and raises the possibility that the vHPC may not play a ubiquitous role in AA conflict processing. Indeed, an absence of anterior HPC involvement has been previously reported in an instrumental AA decision task in non-human primates (*Wallis et al., 2019*). Additionally, we recently reported significant involvement of human PRC rather than HPC

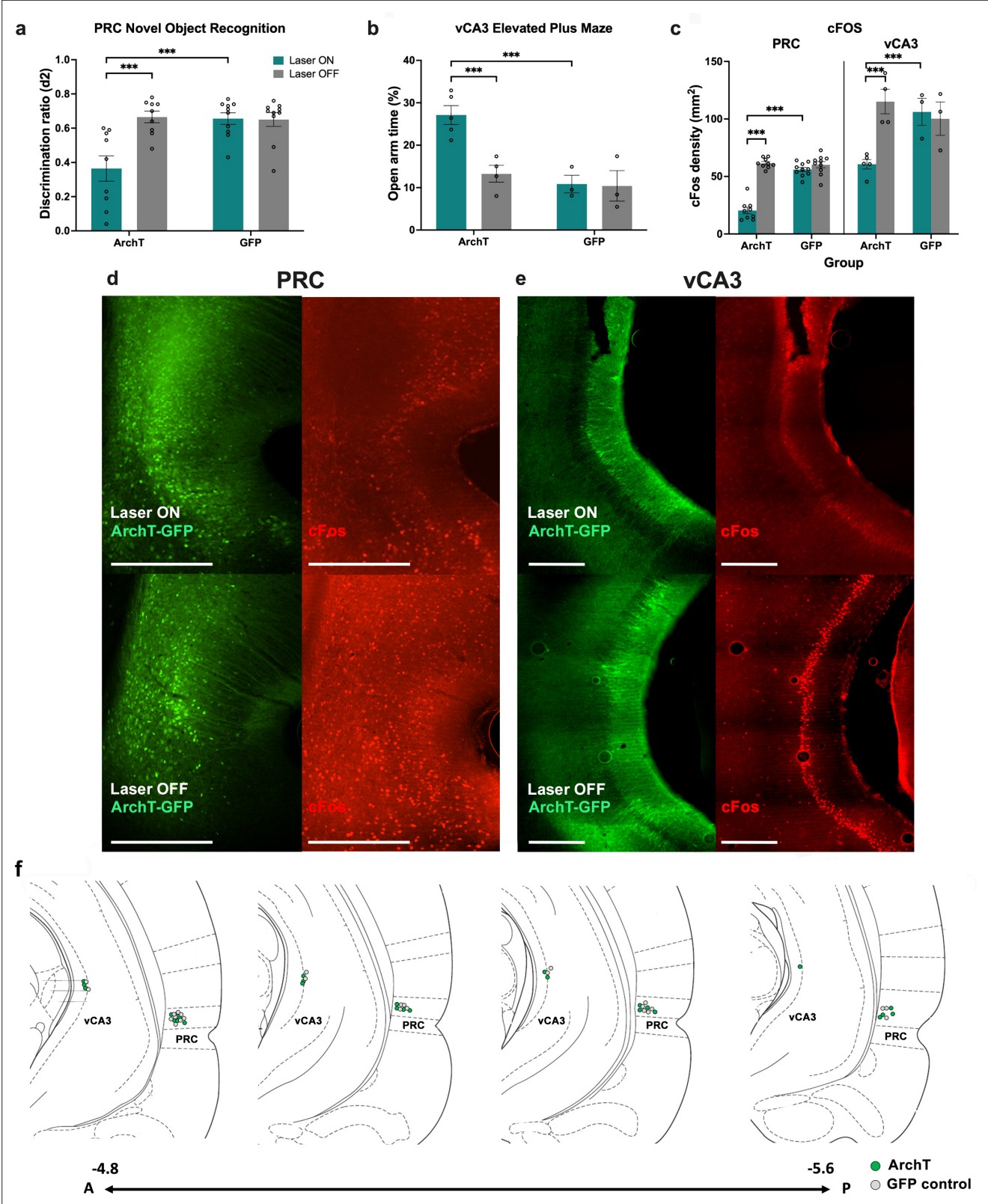

**Figure 11.** Effect of optogenetic inhibition on perirhinal cortex (PRC)- or ventral CA3 (vCA3)-dependent control tasks and cFos expression. (**a**) PRC inhibition disrupted the ability of rats to discriminate a novel and familiar object on the novel object recognition task. (**b**) vCA3 inhibition increased time spent in the open arm of the elevated plus maze. (**c–e**) PRC and vCA3 inhibition was associated with decreased cFos levels in each respective area (PRC: n=18 archaerhodopsin T [ArchT]; n=20green fluorescent protein [GFP], vCA3: n = 9 ArchT, n = 6). (f) Schematic diagram showing placements of optic

*Figure 11 continued on next page*

Figure 11 continued

fibre tips in areas overlying the PRC and vCA3 in sections spanning - 4.8 to -5.6 relative to bregma. Scale bars depict 500 μm. Data figures show mean values ± SEM. Two-way ANOVA was conducted followed by post-hoc tests with Bonferroni correction for all data shown.***p<0.001.

activity, as measured by fMRI, during the resolution of AA conflict induced by objects with opposing valences (*Chu et al., 2021*; *Chu et al., 2023*). The current findings not only provide invaluable reverse-translational evidence for the rodent PRC mediating analogous behavior, but uniquely demonstrate the necessity of PRC to the arbitration of object-associated AA conflict. Importantly, our observation that PRC inhibition did not alter behavior toward a contextual representation of motivational conflict (i.e., conflicting 'context-like' bar cues that span a Y-maze arm) underlines the specificity of the reported PRC effects to object stimuli, and aligns with theoretical viewpoints that posit a degree of stimulus specificity across MTL structures ( *Davachi, 2006*; *Graham et al., 2010*; *Murray et al., 2007*; *Zeidman and Maguire, 2016*).

We suggest that the observed approach bias in the face of object-associated motivational conflict during PRC inhibition reflects both a robust increase in approach behaviour and decrease in avoidance behaviour. The latter is reflected in the fact that PRC inhibition induced an increase in the time spent in the 'Av-Neu low conflict' condition of the runway task and led to a prolonged 'escape latency' when rodents were first exposed to conflicting object pairs. PRC-inhibited rats also spent a greater amount of time in the conflict cued side in the shuttle box task, further lending support to the idea that there was a decrease in avoidance. On the other hand, we also observed that PRC-inhibited animals spent a disproportionate amount of time by the sucrose dispenser in the runway task, seeking out reward under extinction conditions in the face of motivational conflict, which we interpret as an increase in approach behaviour.

Although the observed involvement of the PRC in the arbitration of AA conflict may, at first sight, be unexpected given that this area is traditionally associated with mnemonic function, its involvement in motivational processes is reasonable considering its reciprocal connectivity with limbic centers of the brain, including the amygdala and orbitofrontal, prelimbic and infralimbic cortices (*Tomás Pereira et al., 2016*; *Agster and Burwell, 2009*; *Burwell and Amaral, 1998*; *Tomás Pereira et al., 2016*). The non-human primate PRC has been implicated in reward-related behavior in a role that goes beyond the basic processing of stimulus-valence associations and may be crucial for mediating more nuanced relationships between cues, behavior, and reward outcome (*Suzuki and Naya, 2014*). Specifically, while PRC damage/inactivation does not eliminate reward-related behavior, it does impact an animal's ability to track and flexibly adapt to changes in the magnitude and schedules of reinforcement of expected reward associated with visual stimuli (*Clark et al., 2013*; *Liu et al., 2000*; *Liu and Richmond, 2000*). Moreover, electrophysiological data point toward a role for PRC in processing configurations of multiple visual stimuli to predict upcoming reward (*Ohyama et al., 2012*). Our data complement and extend these findings by suggesting that rodent PRC is not simply necessary for the retrieval of object valence, as evidenced by valence-appropriate discriminative responding for low conflict (i.e., approaching more for appetitive-neutral vs. aversive-neutral), neutral and same valence object pairs during PRC inhibition, and may be engaged as part of the BIS to suppress approach when a violation of expected reward/punishment contingencies (e.g., conflict) is detected. Indeed, the absence of a functional PRC led to animals spending a greater proportion of time by the sucrose dispenser when presented with high conflict stimuli in the runway task, compared with stimuli with low or no motivational conflict, reflecting greater approach in anticipation of impending reward. Similarly, PRC-inhibited animals exhibited increased approach, and decreased avoidance of motivationally conflicting objects in the shuttle box task. Our finding of PRC, but not vHPC involvement in modulating conflict-elicited AA behavior, challenges the perspective proposed by *Gray and McNaughton, 2000* that MTL areas such as the PRC and entorhinal cortex are recruited to provide stimulus information of conflicting goals to the HPC, and that the HPC engages in resolving response conflicts. The present data would suggest that the PRC is an integral component of the BIS when conflicting configurations of learned object stimuli are encountered, independently of HPC recruitment.

An intriguing question is why PRC inhibition potentiated approach, as opposed to avoidance, of an uncertain outcome. In the present study, the excitatory/principal neuronal population (i.e., CaMKIIα-expressing cells) in PRC was targeted and silenced. These neurons receive powerful excitatory inputs from the basolateral amygdala, lateral amygdala, and medial prefrontal cortex (*de Curtis and Paré,*

*2004*; *Sah et al., 2003*; *Smith and Paré, 1994*), which can override a strongly inhibitory intrinsic network within PRC (*Kajiwara and Tominaga, 2021*), and increase the likelihood of signal propagation to the entorhinal cortex and HPC (*Kajiwara et al., 2003*; *Paz and Pare, 2013*). In contrast, stimulation of local, short-range neocortical inputs to PRC have been shown to evoke inhibitory potentials, suggestive of the recruitment of inhibitory interneurons giving rise to feedforward inhibition (*Martina et al., 2001*; *Unal et al., 2013*). Thus, the absence of glutamatergic projection neurons in the PRC could conceivably disrupt the balance of excitation/inhibition in PRC, and lead to a loss of output to a downstream target that is critical in suppressing approach in the face of motivational conflict. In the absence of an effect of inhibiting vHPC, a prominent projection target of PRC, on the expression of object-based AA conflict, we propose the nucleus accumbens, or septum as alternative candidate downstream areas that subserve this function. These areas receive direct projections from PRC (*Tomás Pereira et al., 2016*) and are intrinsically organized (due to GABAergic principal neurons) to inhibit areas further downstream that are concerned with the execution of motor programs and motivated behavior (*Floresco, 2015*; *Wirtshafter and Wilson, 2021*).

Finally, given the well-established role of the PRC in recognition memory and the processing of novelty signals (*Brown and Aggleton, 2001*; *Winters et al., 2008*), it is important to highlight that the present effects of PRC inhibition cannot be explained by a disruption to this function. PRC-intact rats typically increase exploration of novel compared to familiar stimuli (*Winters et al., 2008*) and yet here, PRC inhibition did not change choice behavior during a 'no conflict' test in which novel, recombined stimulus pairs within the same valence were presented. Similarly, it is also unlikely that the observed PRC-mediated impairments stem from a deficit in the perceptual processing of the two objects that comprised the high and low conflict pairs, as evidenced by the ability of PRC-inhibited animals to discriminate between valences during the low conflict test. Furthermore, although there is a delay between stimulus presentation and outcome presentation in the object-based runway AA task, it is unlikely that impairments in trace conditioning, in which the PRC has been implicated (*Kholodar-Smith et al., 2008*), can account for the PRC-mediated impairments in conflict processing. When stimulus presentation occurred in the outcome-associated area in the object-based shuttle box task (i.e., no delay between stimulus and outcome presentation), a similar increase in approach behavior toward high conflict object pairs was observed.

To conclude, we demonstrate that rodent PRC is involved in the resolution of AA conflict, particularly when the goal stimulus is object-based. Thus, stimulus type may fundamentally alter how the brain represents motivational conflict, which may in turn recruit differential MTL structures for the purpose of AA conflict resolution. Our findings also have implications for our understanding of mental disorders in which AA behavior is awry and suggest that a singular focus within the MTL on HPC dysfunction may be inadequate. Further work is needed on delineating the network of structures involved in various aspects of the conflict process, along with examining how a changing stimulus gradient (e.g., object vs. context vs. social) may recruit different brain structures to resolve conflict.

## Materials and methods

### Key resources table

| Reagent type (species) or resource | Designation | Source or reference | Identifiers | Additional information |
|---|---|---|---|---|
| Strain, strain background (*Rattus norvegicus*) | Long-Evans rats, male, 2–6 months old | Charles River | Cat#2308852, RRID:RGD_2308852 | |
| Antibody | Rb Anti-c-Fos rabbit polyclonal | Synaptic Systems | Cat# 226 003, RRID:AB_2231974 | TSA-IHC (1:5000) |
| Antibody | Peroxidase AffiniPure F(ab')$_2$ Fragment Donkey Anti-Rabbit IgG (H+L), donkey polyclonal | Jackson ImmunoResearch | Cat# 711-036-152, RRID: AB_2340590 | TSA-IHC (1:500) |
| Recombinant DNA reagent | pAAV- CaMKIIa-ArchT-GFP | Addgene/Ed Boyden | Cat# 99039-AAV1, RRID: Addgene_99039 | Inhibitory opsin |
| Recombinant DNA reagent | pAAV-CaMKIIa-GFP | Addgene/Bryan Roth | Cat# 50469-AAV8, RRID: Addgene_50469 | Control virus |
| Software, algorithm | SPSS | IBM | https://www.ibm.com/spss | Version 26 |

*Continued on next page*

*Continued*

| Reagent type (species) or resource | Designation | Source or reference | Identifiers | Additional information |
|---|---|---|---|---|
| Software, algorithm | Ethovision XT | Noldus Information Technology | https://www.noldus.com/ethovision-xt | Animal tracking software/ hardware |
| Software, algorithm | Prism | GraphPad | https://www.graphpad.com/ | Version 8 |
| Other | NHS-Rhodamine | Thermo Fisher Scientific | Cat# 46406 | Rhodamine-based dye; TSA-IHC (1:500) |

## Subjects

Subjects were 53 male Long Evans rats (Charles River Laboratories, QC, Canada) weighing between 320 and 380 g at the start of the experiment. Following surgery, rats were individually housed to prevent damage to the implanted optic fiber, and maintained at a constant room temperature of 22°C under a 12 hr light/dark cycle (lights on at 7:00am). Water was available ad libitum, and animals were fed ad libitum up until food-motivated experimental training commenced. Food-restricted animals were maintained at 85–90% of their baseline free-feeding weight. All experiments were conducted during the light cycle and were done in accordance with the regulations of the Canadian Council of Animal Care and approved by the University and Local Animal Care Committee of the University of Toronto (Protocol no. 20012479). We anticipated a moderate to large effect size (0.6–1.1) based on unpublished behavioral data from our laboratory obtained from three previous cohorts of animals. These values were used for an a priori power analysis to compute the required sample size for the current experiment, in which a within-subjects design for optogenetic inhibition was used (*Faul et al., 2007*).

## Surgery

Rats were randomly assigned to receive viral infusions to either the PRC (n=38; n=18 ArchT, n=20 GFP controls) or the vCA3 region of the HPC (n=15; n=9 ArchT, n=6 GFP controls). Rats were anesthetized with isoflurane (Benson Medical, ON, Canada) and secured in a stereotaxic frame (Steolting Co, IL, USA) with the incisor bar set at –3.3 mm below the interaural line. An incision along the midline of the skull was made and the fascia retracted by small skin clips to reveal bregma. For PRC surgeries, the temporalis muscle was carefully peeled back from the temporal ridge to access lateral injection sites. Small burr holes were created directly over the injection sites using a dental drill, and 0.5 µl of virus (either AAV1-CaMKIIa-ArchT-GFP or AAV8-CaMKIIa-GFP, Addgene, MA, USA) was infused bilaterally with a 1 µl Hamilton syringe into either the PRC (AP: –5.2 mm, ML: ±6.7 mm, DV: –7.3 mm) or the vCA3 (AP: –5.2 mm, ML: ±4.8 mm, DV: –7.0 mm) over 5 min, with the needle left in situ for a further 5 min. Optic fibers were then implanted 0.5 mm dorsal to the infusion site, and were secured in place with dental cement anchored to jeweler screws implanted in the skull. Rats were given at least 7 days to recover in their home-cages before beginning behavioral experiments. The viral infusion and optic fiber implantation coordinates were chosen based on two previous cohorts of animals (*unpublished observations*) that demonstrated refined viral expression patterns and optimal fiber implantation localized to the PRC and vCA3 regions.

## Apparatuses
### Object-based AA task apparatus

Testing was conducted in three Y-maze arms (each 77 cm [L] × 11.5 cm [W] × 35 cm [H]) arranged to form a continuous 'runway' (231 cm [L]), with the sides of the apparatus were wrapped in red cellophane to minimize reliance on extra-maze cues, with the top remaining exposed to permit optogenetic tethering and free mobility of the compartment door. The runway was segmented into four compartments (start box, object box, neutral box, and goal box) which were separable by stainless-steel guillotine door inserts. The entire stainless-steel grid floor was covered in opaque black Plexiglas, except for the goal box, which contained exposed grid flooring connected to a shock generator (Med Associates, VT, USA). A stainless-steel dish connected to a sucrose dispenser was presented at the end of the grid flooring (in the goal box), which the animals were required to traverse in order to obtain reward.

## Object-based shuttle box apparatus

Behavioral testing was conducted in a modified active avoidance 'shuttle box' apparatus (54 cm [L] × 26 cm [W] × 32.5 cm [H]) (Coulbourn Instruments, Whitehall, PA, USA), which was divided into two equally halved compartments (27 cm [L] × 26 cm [W] × 32.5 cm [H]) separated by a central guillotine door, with the entire apparatus wrapped in red cellophane to minimize reliance on extra-maze cues. Two removable panels (40 cm [L] × 25 cm [W]), one opaque and one transparent, were inserted in each side of the apparatus, which served to prevent recognition of and access to the object pairs, respectively. The transparent panel contained two rows of holes (each 4 cm apart), positioned at the rat's eye level, to permit olfactory sampling. Two stainless-steel dishes connected to sucrose dispensers were presented at opposing ends of the apparatus, located behind both panels.

## Y-maze AA task apparatus

Behavioral testing was conducted in a three-arm Y-maze apparatus, as previously described (*Nguyen et al., 2019*; *Schumacher et al., 2018*; *Schumacher et al., 2016*; *Yeates et al., 2020*). Briefly, each arm (50 cm [L] × 11.5 cm [W] × 35 cm [H]) was connected to a hexagonal central hub compartment (11.5 cm [W] × 35 cm [H]), with each arm wrapped in red cellophane to minimize reliance on extra-maze cues and arranged 120° relative to the adjacent arm. Arm entrances were blocked by stainless-steel guillotine doors. The grid flooring for each arm was connected to a shock generator (Med Associates, VT, USA) and led to a stainless-steel dish connected to a sucrose dispenser.

## NOR apparatus

Behavioral testing was conducted in a transparent open-field apparatus (50 cm [L] × 50 cm [W] × 50 cm [H]), wrapped with black plastic to reduce anxiety. The apparatus was lined with home-cage bedding, which was replaced on days of testing. The bedding was agitated after each trial to eliminate potential odor traces.

## Stimuli

### Object cues

Given the established role for the PRC in the resolution of feature ambiguity between highly similar objects (*Saksida and Bussey, 2010*), we attempted to maximize the similarity between selected objects by restricting the height (from 3.5 to 20 cm) and composition (either glass, plastic, or metal) of the object pair. A collection of 'junk objects' (i.e., no prior reinforcement history and no natural significance to the rats) were obtained for behavioral testing for the three behavioral experiments (object-based runway task, object-based shuttle box task, NOR). For NOR task, the pairs of objects were composed of the same material so they could not be readily discriminated by olfactory cues, whereas object exploration preference was used to assign pairings for the object-based AA tasks (see task procedure below). The objects were attached to the apparatus floor with a hook-and-loop fastener. All objects were cleaned with 70% ethanol solution after each trial for all experiments.

### Bar cues

A set of visuotactile cues were used in the Y-maze task, which consisted of wood panel inserts (46.5 cm [L] × 9.6 cm [H]) affixed to the length of each arm with hook-and-loop fasteners. Three sets of bar cues were used, with two sets wrapped in either duct tape or a denim cloth material, and the last set being an unwrapped, varnished wooden material. Bar cues were wiped down with a 70% ethanol solution between trials.

## Behavioral procedures

PRC rats and VCA3 rats first completed testing in the object-based AA runway task (*Figure 1*). They then underwent testing in the object-based AA shuttle box task. The PRC rats were then trained in the RAM AA task, prior to completing a final NOR test before euthanasia. vCA3 rats were administered a final EPM task before euthanasia.

## Object-based AA runway task

### Habituation

All rats were given 4 days of habituation to permit exposure to the runway apparatus and the object cues. On day 1, rats were confined to the start box for 30 s, and then permitted to explore the entire apparatus for 3.5 min. On day 2, following a 30 s confinement to the start box, rats could proceed into and were confined within the object box (without any objects) for 1 min, with re-entry to the start box prohibited by a stainless-steel guillotine door. Rats could then proceed to the neutral box, in which they were confined for 1 min, with re-entry to the object boxes prohibited. Following this, they were confined to the goal box for 30 s before being removed from the apparatus. Days 3 and 4 introduced the rats to the six object cues used for the object-based AA task; each rat performed three daily trials, with each trial presenting two object cues, and thus rats were exposed to all six object cues daily. The order that the objects appeared for each rat was counterbalanced between animals and habituation sessions, and the time spent exploring the objects was recorded. Object habituation began with 30 s confinement to the start box, and rats were then permitted 1 min to explore the two object cues during a given session. Rats then entered the neutral box, in which they were confined for 1 min, followed by the goal box, in which they were confined for 30 s. The affective valence of the object cues for subsequent conditioning sessions was determined based on exploration latencies during habituation 3 and 4. The two most-explored object cues were assigned as the aversive cue pair, the two least-explored as the appetitive cue pair, and the remaining two as the neutral cue pair.

### Object cue-outcome conditioning

The animals were trained to associate the three object pairs with either appetitive (sucrose), aversive (mild foot shock), or neutral (no event) outcomes over nine daily conditioning sessions. Each rat completed three trials per day, one for each affective valence. During a trial, the time spent exploring each object cue was recorded, and the rat could enter the neutral box after 1 min elapsed. After 10 s, rats could enter the goal box where they were confined to for 30 s during which the outcome assigned to the explored object pair was administered (appetitive: 2×0.8 ml of 20% sucrose every 15 s; aversive: 2×0.75 s, 0.26–0.29 mA shock every 15 s; neutral: no outcome). The magnitude of shock was deliberately kept at a level to elicit avoidance behaviour but preserve the animals' exploratory drive (i.e., minimize freezing). After 30 s elapsed, rats were removed from the apparatus and returned to their home-cage in preparation for the next conditioning trial. If rats did not readily enter the next box within 30 s, they were gently ushered in by the experimenter, and object, neutral, and goal boxes LTE data was also recorded as a potential measure of preference/aversion of expected outcome. The shock level was calibrated for each rat during the first aversive conditioning session and fixed at a level which elicited a mild startle response and defensive treading behavior, but not freezing. The order of presentation of the object cues and the order in which rats completed each trial were changed daily.

### Conditioned cue acquisition test

Acquisition tests performed under extinction conditions were conducted after conditioning days 4 and 8 in order to assess learning of the object cue-outcome associations. The experimental procedure was identical to that of cue conditioning training, except that rats were not confined to the goal box upon entry and were permitted 2 min to freely move between the neutral box and goal box during this time. Successful acquisition was indicated by the rats spending more time in the goal box for appetitive trials than aversive and neutral trials, and rats spending more time in the neutral box for aversive trials than appetitive and neutral trials. Following the second acquisition test, rats were given a 'refresher' conditioning day prior to proceeding to the AA conflict test.

### High AA conflict test

On the day of AA conflict testing, rats were bilaterally tethered to the laser and placed into the start box. The laser was then turned on for animals completing the entire task under PRC inhibition, before the start box guillotine door was removed exposing the animals to the object box, to which they were confined upon entry. One appetitive object cue and one aversive object cue were presented in recombination as the 'high conflict object pair', which animals could freely explore for 1 min. Identical to conditioning sessions, rats were then given 30 s to freely enter the neutral box, after which they were gently ushered in by the experimenter, and confined therein for 10 s. Rats were then permitted

to enter the goal box. At this point, rats were given 3 min to freely move between the neutral box and goal box, after which the laser was turned off and the animal was removed from the apparatus and returned to its home-cage in preparation for the next trial. All rats completed two conflict test sessions, one with the laser on and one with the laser off, each with a different set of conflict object cues. The order of the laser treatment, the assigned combination of the conflict object cues, and whether the aversive or appetitive object was presented first in the runway were all counterbalanced.

The time spent in the neutral and goal boxes, the number of entries into the goal box, and LTE each of the boxes (object, neutral, goal) in the runway were recorded during the tests. As an additional measure of AA behavior, the number of full-body entries into and retreats from the goal box were recorded. A full-body entry occurred when the animal's hind limbs stepped on to the grid floor of the goal box. To be consistent with our previous work using the same Y-maze task (*Schumacher et al., 2018*), a retreat occurred when the animal exhibited a half-body entry (e.g., only forelimbs) or head-poke into the goal box followed by an immediate exit or backward treading into the neutral box.

### Neutral object test

To ensure that PRC-inhibited could still discriminate stimuli that they were trained on during the conditioning session, a test with the neutral object pair was conducted on the same day as the two AA conflict tests, during which half of the rats (PRC n=19 [n=9 ArchT]; vCA3 n=8 [n=5 ArchT]) completed the test with the laser on, and half with the laser off. . Unlike the high approach-avoidance conflict test, the neutral object test was conducted in a between-subjects design since we deemed that a fully counterbalanced experimental design with 2 conflict trials and 2 neutral trials might lead to unreliable data during the last trial owing to extinction-related confounds. We therefore prioritized maximizing the reliability of the data collected from the 2 conflict trials. The animals were allocated into the Laser on/off groups randomly, after it was confirmed that baseline performance during the conflict trials was comparable across animals.

### Low AA conflict tests

To investigate whether reducing the level of object-associated motivational conflict might change PRC-mediated behavior, both PRC and vCA3 rats were given a 'refresher' conditioning session prior to administration of a 'low conflict' recombination test, during which either an appetitive or aversive object cue was paired with a neutral object cue. PRC and vCA3 rats were each given two trials per test, an appetitive-neutral object pairing, and an aversive-neutral pairing. Each rat was tested twice, once with the laser on throughout the entire test session, and once with the laser off. The order of laser treatment, the assigned combination of the object cues, and whether the appetitive/aversive or neutral object was presented first were counterbalanced within and between each test session.

In a separate cohort of animals (n=16; n=8 ArchT), all AA conflict tests (high and low) and neutral object test were administered exactly as described above, with the exception that the laser was turned on after the animals entered the neutral box, and prior to the door to the goal box being raised (i.e., choice period, 10s). The laser then remained on for the duration of the goal box exploration (3min).

### No conflict tests

Finally, to control for the novelty of the recombined object pair presented during both the 'high' and 'low' conflict tests, along with an additional measure to confirm that PRC rats could still discriminate stimuli under optogenetic inhibition, all rats (except for PRC ArchT-expressing rats undergoing pre-choice inhibition) completed a 'no conflict' test, during which a novel recombination of objects of the same valence were presented. Rats were first trained to associate a novel set of eight objects with either appetitive (four objects; two object pairs) or aversive outcome across four daily conditioning sessions, followed by an acquisition test and a 'refresher' conditioning session prior to testing. On the day of testing, rats (PRC n=16 [n=8 ArchT]; vCA3 n=8 [n=5 ArchT]) were given two trials per test, an appetitive-appetitive object pairing, and an aversive-aversive pairing. Each rat was tested twice, once with the laser on throughout the entire test session, and once with the laser off.

## Object-based AA shuttle box task

### Habituation

All rats were given 3 days of habituation the runway apparatus. On day 1, rats could freely explore both sides of the apparatus for 5 min, with the central guillotine door raised. On day 2, the rats were confined to one side of the apparatus with the guillotine door and following 2.5 min of exploration, the door was raised and animals were permitted to shuttle to the opposite side of the apparatus. Upon entry, the door was lowered and the animals were confined for a further 2.5 min. On day 3, the rats were confined to one side of the apparatus which now contained two sets of removable panels – one opaque and one transparent, preventing access to the sucrose dishes located at the end of the two compartments. After 10 s the opaque panel was removed, providing visual but not tactile access to the sucrose dispenser. After a further 30 s, the transparent panel was removed and the central guillotine door was simultaneously raised. The animals could then freely explore the empty sucrose dish, and were permitted to shuttle to the opposite side, which contained another set of opaque and transparent panels. Once animals shuttled to the opposite side, the door was lowered, and the habituation procedure was repeated in the other compartment.

### Object cue-outcome conditioning

The animals were trained to associate the two object pairs with either appetitive (sucrose) or aversive (mild foot shock) outcomes over nine daily conditioning sessions. Each rat completed four trials per day, two for each valence. During a trial, rats were placed between the central guillotine door and the opaque panel, confining them to one side of the apparatus. Following 10 s the opaque panel was removed and the animals were permitted to visually sample the object pair for 1 min, which remained behind the transparent panel. After 1 min elapsed, the transparent barrier was removed, the central door was simultaneously raised, and the outcome delivered (appetitive: 1×1.5 ml of 20% sucrose; aversive: 1 s 0.24–0.27 mA shock, pulsed every 2 s). Animals were given 1 min to shuttle to the opposite side of the apparatus before they were gently ushered in by the experimenter. Upon shuttling, the central door was lowered and the next trial began. The time spent visually sampling the objects, defined as the amount of time the rats directed their nose toward the object immediately behind the transparent panel, and escape latency data were collected. The order of presentation of the object cues and the order in which rats completed each trial were changed daily.

### Conditioned cue acquisition test

Acquisition tests performed under extinction conditions were administered after conditioning days 4 and 8 to assess learning of the object cue-outcome associations. The experimental procedure was identical to that of cue conditioning training, except that the central guillotine door was not lowered upon the first shuttle response, and rats were permitted to re-enter the cued side of the apparatus and given 3 min to freely move between the cued and uncued sides. Similar to conditioning sessions, the uncued side contained an opaque panel. Successful acquisition was indicated by the rats exhibiting significantly shorter escape latencies for aversive trials than appetitive trials, and spending less time in the cued side during aversive trials than appetitive trials. Following the second acquisition test, rats were given a 'refresher' conditioning day prior to proceeding to the conflict test.

### AA conflict test

On the day of AA conflict testing, rats were bilaterally tethered to the laser and rats were placed between the central guillotine door and the opaque panel, confining them to one side of the apparatus. The laser was then turned on, and remained illuminated for the entire duration of the test. The opaque panel was then removed after 10 s, exposing the animals to the object pairs behind the transparent panel. One appetitive object cue and one aversive object cue were presented in recombination as the 'conflict object pair', which animals could visually sample for 1 min. Once 1 min had elapsed, the transparent panel was removed and the central door was simultaneously raised. At this point, rats were given 3 min to freely move between the 'conflict-cued' side of the apparatus and the uncued side, after which the animal was gently ushered into and confined to the uncued side by the experimenter. The laser was then turned off. All rats completed two conflict test sessions under extinction conditions, one with the laser on and one with the laser off, each with a different set of conflict object cues. The escape latency and time spent in the 'conflict-cued' side of the apparatus were recorded,

along with the number of re-entries into the cued side; a re-entry was defined when the animal's hind limbs crossed the threshold of the central door of the cued side. The order of the laser treatment and the assigned combination of the conflict object cues were all counterbalanced.

In a separate cohort of animals (n=16; n=8 ArchT), the conflict test was administered exactly as described above, with the exception that the laser was turned on as the transparent door was removed after visual inspection of the object pair. The laser then remained on for the duration of the object exploration (3 min).

## Conditioned cue acquisition tests with PRC/vCA3 manipulation

Following the conflict test, animals were given a 'refresher conditioning' session, prior to completing two sets of conditioned cue acquisition tests, one with the laser on and one with the laser off. The procedure of the tests was identical to the conditioned cue acquisition test described above, which were conducted under extinction conditions. On the day of testing, the laser was turned on for animals completing the task under inactivation conditions before completing one appetitive and one aversive acquisition test. On the next day all rats were given another 'refresher conditioning' session before completing another set of acquisition tests the following day, with the laser turned off for the animals completing the task without inactivation conditions.

## Learned AA Y-maze task

### Habituation

PRC rats underwent four, 5 min habituation sessions as previously described (*Nguyen et al., 2019*; *Schumacher et al., 2018*). Briefly, on day 1, following 1 min confinement to the central hub, all guillotine doors were opened allowing 5 min of free exploration of the three arms without cues present. On day 2, a different set of cue inserts were placed into each arm and rats could freely explore the cues for 5 min. The affective valence of the cues for subsequent conditioning sessions was determined based on exploration time during this session. The most-explored cue was assigned as the aversive cue, the least-explored as the appetitive cue, and the remaining one as the neutral cue. On day 3, two guillotine doors were lifted and rats could explore the newly assigned neutral cue in one arm, and a superimposition of the appetitive and aversive cues in the other arm for 5 min; this would mirror the conditions for the final conflict test. The final habituation session was done without cues and habituated rats to confinement within the arms. Following 1 min in the hub, the rats would sequentially enter each arm, and were confined to them for 1 min before being allowed to return to the central hub.

### Cue-outcome conditioning

PRC rats were trained to associate three sets of visuotactile cues with appetitive (sucrose), aversive (shock), or neutral (no event) outcomes over nine daily conditioning sessions. The rat was placed in the central hub for 30 s after which one guillotine door was raised permitting entry to an arm, followed by confinement in the arm for 2 min. During this time the outcome assigned to the cue was administered (appetitive: 4×0.4 ml of 20% sucrose administered every 30 s; aversive: 4×0.75 s, 0.26–0.29 mA shock delivered at a random point every 30 s; neutral: no outcome). After 2 min elapsed, the door was raised and rats were permitted to return to the central hub, and this process was repeated for the remaining two arms. The assignment of each cue to a given arm and the order in which each cue and arm were presented was changed daily. While the relative shape of the maze was held constant between sessions (Y-maze configuration), the maze was rotated either clockwise or counter-clockwise by 60° for each training session to prevent the use of spatial cues.

### Conditioned cue acquisition test

Acquisition tests performed under extinction conditions were conducted after the fourth and eighth training sessions. The test procedure was identical to day 2 of habituation in that rats were given 5 min to explore all three arms simultaneously, but with cue inserts under extinction conditions. Successful acquisition of conditioned behavior was indicated by the rats spending more time in the appetitive arm than aversive and neutral arms, and rats spending less time in the aversive arm than the appetitive and neutral arms. Following the second acquisition test, rats were given a 'refresher' conditioning session prior to proceeding to the AA conflict test.

## AA conflict test

The procedure for this test was identical to day 3 of habituation, with rats being permitted 3 min under extinction conditions to explore two arms presented simultaneously, one containing the neutral cue and one containing a superimposition of the appetitive and aversive cues. In addition to the time spent in each arm, the number of entries and retreats into both arms was also recorded. All rats completed two test sessions on the same day, one with the laser on for the entire 3 min test period, and one with the laser off. The order of laser treatment and the assignment of the 'conflict' and neutral cues to a given arm were counterbalanced within and between test sessions.

## Laser delivery

Five hundred and thirty-two nm (green) laser light was continuously applied for the length of time specified in each experiment. Laser illumination was delivered to an implanted optic fiber attached with plastic sleeves (diameter 2.5 mm; Doric Lenses, QC, Canada) via two steel-braided fiber-optic cables (200 µm core; 0.22NA; Doric Lenses), which received light through a bifurcating rotary joint (Doric Lenses) secured with an FC connector. The rotary joint was connected to a class 3-B diode-pumped solid-state laser (165 mW output; Laserglow Technologies, ON, Canada) via an FC connector. The light output of the optic fiber was adjusted to approximately 15 mW, and based on previous measurements (*Deisseroth, 2012*) incorporating geometric loss of light, this would produce an irradiance of 15.38 mW/mm$^2$ (0.22NA; fiber core radius = 200 µm; implanted 0.5 mm above target site), which exceeds the minimum amount needed to produce opsin activation (*Gradinaru et al., 2009*; *Stefanik et al., 2013*; *Tye et al., 2011*).

## cFos activation and histology

Prior to sacrifice, rats either completed a NOR task (PRC n=38) or an EPM test (vCA3 n=15) to endogenously increase c-Fos labeling in the respective brain areas. Previous work has demonstrated that c-Fos is consistently increased in the rat PRC following exposure to novel visual stimuli (*Albasser et al., 2010*; *Zhu et al., 1995*). Similarly, rats exposed to anxiety-provoking environments (e.g., EPM) consistently demonstrate increased c-Fos labeling in the vHPC (*Duncan et al., 1996*; *Hale et al., 2008*; *Linden et al., 2004*).

The NOR task was conducted in the open-field apparatus (50 cm [L] × 50 cm [W] × 50 cm [H]). Rats were first given two daily 5 min habituation sessions of the apparatus, without any object stimuli. On the day of testing, the bedding in the apparatus was changed, and rats were placed in the open-field facing away from two identical objects (A1 and A2), which they were permitted to 'sample' for 5 min. Rats were subsequently returned to their home-cage for a 15 min delay interval. During the test phase, rats were returned to the open field, which now contained a third identical copy of the sample object (A3) and a novel object (B1), which they were permitted to explore for 3 min. Between the sample and test phases, the apparatus was not cleaned, nor the bedding agitated, and no other rats were placed in the apparatus during this time. Half of the rats received laser treatment (n=19 [ArchT = 9]) during the sample and test phases, and half completed the task with the laser off.

The EPM apparatus consisted of a central area (10 cm [L] × 10 cm [W]) with four maze arms (43.2 [L] × 10 [W] × 43.2 [H]) forming a plus shape. Two maze arms, directly across from one another, were enclosed by high walls (24.8 cm [H]), while the other two arms were unenclosed or 'open'. Rats were placed into the central area facing the 'open' arm, and were given 10 min to freely explore the apparatus. Half of the rats (n=8 [ArchT = 5]) completed the task with the laser on, while the other half with the laser off.

Ninety minutes after completing the test phase in the NOR task or the EPM test, rats were administered a terminal dose of sodium pentobarbital (Bimeda-MTC, Cambridge, ON) and were intracardially perfused with phosphate-buffered saline (PBS), followed by 4% paraformaldehyde (PFA). Brains were removed and stored in 4% PFA for 24 hr before sectioning. Fifty µm coronal sections were obtained with a vibratome (VT1000s; Leica Microsystems, Germany), mounted onto glass slides with a Flouroshield mounting medium containing DAPI (Ab104139; Abcam). Optic fiber placement and viral injection site targeting were confirmed at ×10 and ×20 magnification under an Eclipse Ni-U epifluorescence microscope (Nikon Instruments, Japan) using a FITC filter.

For cFos immunoreactivity, sections were washed (five times for 5 min with PBS on a shaker) and incubated in 1% H$_2$O$_2$ in PBS for 30 min. Sections were then incubated in 0.5% TNB blocking buffer

for 1 hr, followed by rabbit anti-cFos (in 0.5% TNB; 1:5000; Synaptic Systems), and left overnight at 4°C on a shaker. The next day, sections were first incubated for 1 hr in donkey anti-rabbit (conjugated with horseradish peroxidase in 0.5% TNB; 1:500; Jackson ImmunoResearch), and then in diluted Rhodamine tyramide signal amplification (TSA) solution (in 0.01% $H_2O_2$ in 0.1 M borate buffer; 1:500; Jackson ImmunoResearch) and wrapped in foil to prevent photobleaching for 30 min. Sections were mounted onto glass slides and treated with a mounting medium (Flouroshield, Ab104139; Abcam). Labeling of c-Fos proteins in the PRC and vCA3 were confirmed at ×10 magnification under the Eclipse Ni-U epifluorescence microscope (Nikon Instruments, Japan) using FITC and TexasRed filters, with cell counting performed by an automated counting software (Fiji; *Schindelin et al., 2012*). cFos quantification was performed bilaterally, spanning three separate sections of tissue for both the PRC and vCA3 (AP: from –4.8 to –5.6).

## Data analysis

All behavioral testing was recorded and tracked using Noldus EthoVision XT (Noldus, Netherlands). Data were analyzed using SPSS statistical package version 26.0 (IBM, ON, Canada). For the object-based AA task, conditioned cue acquisition test data for the time spent in the goal box, the LTE, number of entries and retreats, and object exploration latencies were analyzed by separate 3×2 repeated-measures ANOVA with a within-subjects factor of valence (appetitive, aversive, neutral), and a between-subjects factor of group. Analyzed data for the high conflict test were identical to the acquisition test (including object exploration data) and were subject to a 2×2 repeated-measures ANOVA with within-subjects factors of laser treatment, and a between-subjects factor of group. Each rat only completed one session of the neutral test, with between-subjects factors of laser treatment and group, and data were analyzed by univariate ANOVA. Data for the low conflict and no conflict control tests were each first analyzed by separate 2×2×2 repeated-measures ANOVA with within-subjects factors of valence and laser treatment, and a between-subjects factor of group. Activity analysis by the sucrose dispenser and generation of heatmap figures were completed using Noldus EthoVision XT, in which an 18 cm (L) × 11.5 cm (W) region of interest was demarcated within the goal box (around the dispenser; 1/3 of the total goal box), and could reliably contain an entire rat.

For the Y-maze AA task, conditioned cue acquisition data for the time spent in each of the cued arms, as well as the number of entries and retreats were analyzed by a 3×2 repeated-measures ANOVA with a within-subjects factor of valence (appetitive, aversive, neutral) and a between-subjects factor of group. Analyzed data for the conflict test were identical to the acquisition test and were subject to a 2×2×2 repeated-measures ANOVA with within-subjects factors of laser treatment and trial type (conflict vs neutral), and a between-subjects factor of group. For the EPM test data, 2 x 2×2 repeated-measure ANOVAs were conducted to compare the time spent in the open and closed arms, as well as the number of entries made into both, with a within-subjects factor of arm type (open vs closed) and between-subjects factors of laser treatment and group.

For the NOR task, a 2×2 repeated-measures ANOVA was conducted on the 'discrimination ratio' ($d_2 = \frac{(t)\,\text{Novel} - (t)\,\text{Familiar}}{(t)\,\text{Total}}$), with a within-subjects factor of laser treatment and between-subjects factor of group. Behavioral data from the NOR task revealed that consistent with previous findings (*Dix and Aggleton, 1999*; *Winters et al., 2004*; *Winters and Bussey, 2005*), both novel object exploration and novel object discrimination performance were significantly higher during the first minute of the test session (time point: $F_{(2,28)}=20.79$, $p<0.001$; object × time point: $F_{(2,28)}=16.41$, $p<0.001$), and thus, this time point was selected for subsequent analysis. For cFos quantification, a 2×2 ANOVA was conducted on the density of cFos labeled cells in both the PRC and vCA3, with between-subjects factors of laser treatment (on vs off) and group.

Significant main effects and interactions were further investigated with simple contrasts and post hoc tests with a Bonferroni correction where appropriate. Violations to sphericity were addressed with a Huynh-Feldt correction. All manually scored data (e.g., entries, time spent in a given zone) were blindly scored by a second experimenter. None of the data collected were excluded in the final analysis.

## Acknowledgements

This work was funded by the Canadian Institutes of Health Research (#156070 to ACHL and RI). The authors gratefully acknowledge Dr Maithe Arruda-Carvalho and the members of her DevNeuro lab in allowing us to use their Ni-U epifluorescence microscope.

## Additional information

### Funding

| Funder | Grant reference number | Author |
|---|---|---|
| Canadian Institutes of Health Research | 156070 | Andy CH Lee<br>Rutsuko Ito |

The funders had no role in study design, data collection and interpretation, or the decision to submit the work for publication.

### Author contributions

Sandeep S Dhawan, Formal analysis, Investigation, Visualization, Methodology, Writing – original draft, Writing – review and editing; Carl Pinter, Investigation; Andy CH Lee, Conceptualization, Supervision, Funding acquisition, Visualization, Writing – original draft, Writing – review and editing, Methodology; Rutsuko Ito, Conceptualization, Supervision, Funding acquisition, Visualization, Methodology, Writing – original draft, Writing – review and editing

### Author ORCIDs

Sandeep S Dhawan ⓘ http://orcid.org/0000-0002-6511-7678
Andy CH Lee ⓘ http://orcid.org/0000-0002-8546-5311
Rutsuko Ito ⓘ http://orcid.org/0000-0003-1769-5470

### Ethics

All experiments were conducted in accordance with the regulations of the Canadian Council of Animal Care and approved by the University and Local Animal Care Committee of the University of Toronto (Protocol no. 20012479).

### Decision letter and Author response

Decision letter https://doi.org/10.7554/eLife.81467.sa1
Author response https://doi.org/10.7554/eLife.81467.sa2

## Additional files

### Supplementary files
• MDAR checklist

### Data availability
All data generated in this study have been deposited in Open Science Framework database under the accession code: https://osf.io/9h7wr/.

The following dataset was generated:

| Author(s) | Year | Dataset title | Dataset URL | Database and Identifier |
|---|---|---|---|---|
| Dhawan S, Lee ACH, Ito R | 2021 | Male rodent perirhinal cortex, but not ventral hippocampus, inhibition induces approach bias under object-based approach-avoidance conflict | https://osf.io/9h7wr/ | Open Science Framework, OSF |

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
