## [Editor Report]

In this important study the authors combined innovative object-based conflict assays with optogenetic silencing to probe the role of the perirhinal cortex in motivational conflict. The evidence provided by the authors is convincing and provides new insight into how conflicting motivation is processed. This paper will interest neuroscientists studying the neuronal mechanisms underlying approach-avoidance decisions.

---

## [Decision Letter]

**Decision letter after peer review:**

Thank you for submitting your article "Rodent perirhinal cortex, but not ventral hippocampus, inhibition induces approach bias under object-based approach-avoidance conflict" for consideration by *eLife*. Your article has been reviewed by 3 peer reviewers, and the evaluation has been overseen by a Reviewing Editor and Kate Wassum as the Senior Editor. The following individuals involved in the review of your submission have agreed to reveal their identity: Christian Bravo-Rivera (Reviewer #2); Lukas I Schmitt (Reviewer #3).

Essential revisions:

*Reviewer #1 (Recommendations for the authors):*

General

1. Inhibition of the PRC altered conflict processing in the first and second tasks (runway and shuttle box, respectively) but not in the third (Y-maze). How do we know that the failure to detect an effect of PRC inhibition in the third conflict task is due to the fact that it requires context/spatial processing as opposed to reduced efficacy of the repeated PRC inhibition?

2. How did animals interact with the object pairs in conflict trials? That is, did they preferentially explore one or the other object, and did PRC inhibition alter this exploration in a manner that correlates with its impact on test performance? I.e., did it create a bias towards the exploration of the object that signaled the appetitive outcome? The authors indicate that the exploration data were recorded; their inclusion in the manuscript could help to clarify how inhibition of the PRC has the effects that it does in the conflict test.

3a. L151. "When animals were exposed to a high conflict pairing, optogenetic inhibition of PRC (laser on) significantly increased time spent in the goal box compared with trials completed without inhibition (laser off) and compared to GFP control animals with laser on and off (laser x construct: F(1, 16) = 29.03, p < 0.001, ηp2 = 0.65; all post hoc: p < 0.001) (Figure 2a), indicative of an increase in approach behaviour under motivational conflict. Furthermore, PRC inhibition led to a significant decrease in the number of retreats from the goal box (laser x construct: F(1,16) = 16.80, p < 0.001, ηp2 = 0.51), although there was no effect on the number of goal box entries (laser x construct: F(1,16) = 1.92, p = 0.19, ηp2 = 0.11) (Figure 2b) during the choice period." Does the fact that PRC inhibited rats stay longer in the goal box indicate an increase in approach behaviour? These rats enter the goal box just as readily as the controls (entry and latency data) but stay longer when neither outcome occurs; so perhaps they are less sensitive to the absence of sucrose than the controls? The fact that PRC inhibition also increased time in the goal box on trials that included aversive-neutral object pairs (~L169) suggests that it may generally reduce avoidance of an aversive outcome. Is this a fair description of the conflict results? That is, PRC inhibition may have resulted in an appetitive response bias by reducing sensitivity to (or decreasing avoidance of) an aversive outcome…

3b. L490. "Using a set of original object-based AA paradigms, we found that optogenetic inhibition of the rodent PRC, but not the vCA3, resulted in both a robust increase in approach behaviour and decrease in avoidance behaviour during motivational conflict elicited by the representation of discrete object pairs associated with unmatching affective values." What is the evidence of an increase in appetitive behaviour and decrease in avoidance behaviour?

Design

1. How were the rats allocated to groups for the Neu-Neu test shown in Figure 2c? Why was this testing conducted between-subjects (resulting in ns of just 5 per group) rather than within-subjects, as in the previous high conflict test? Did the 5 rats in each ArchT group (laser on vs laser off) and each GFP group (laser on vs laser off) exhibit the same profile of responding in the previous high conflict test? An obvious concern is that the allocation of rats to groups biased against detection of differences among the ArchT groups.

2a. Why are there so few rats in the "Low Conflict-GFP" conditions? That is, why weren't rats in the "High Conflict-GFP" condition carried forward to these additional tests? It renders analyses of the Low and NO conflict conditions problematic: as there are 8 rats in the ArchT group and 4 rats in the GFP group, how can these be submitted to any sort of common analysis via ANOVA?

2b. How many rats were in the ArchT and GFP groups for the novel recombination tests in the runway experiment? The figure again suggests 8 and 4, which again renders the ANOVA problematic.

2c. The query in relation to unequal ns and application of ANOVA also applies to the results of the shuttle box experiment shown in Figure 3f-i (n for ArchT = 7; n for GFP = 4); and to the combination of PRC and vHPC data that is reported to have been analysed together near line 379. Can this be legitimized in some way?

3. L640, 652, 664. How were rats tethered to the lasering in the AA conflict test as "the entire apparatus [was] wrapped in red cellophane to minimize reliance on extra-maze cues"?

Data/stats

1. L551: "An intriguing question is why PRC inhibition potentiated approach, as opposed to avoidance, of an uncertain outcome." I understand that there is no effect of the laser on latency to enter any of the runway sections in the high and low conflict tasks. I assume that this is due to the fact that latency data is notoriously variable and the ns are relatively low. As such, I would be interested in an assessment of whether there is an effect of the laser on latency to enter any of the runway sections in an analysis that combines high and low conflict tasks that include an appetitive component…

2. In task 2, are the escape latency and "time in cued side" data correlated? If so, they are potentially two measures of the same thing, which would warrant some correction in the α rate to control the chance of a Type 1 error. This may have consequences for claims based on these data (e.g., analysis of data presented in Figure 3g).

3. L765. "The number of entries into and 'retreats' from the outcome box were recorded; an entry was defined when the animal's hind limbs stepped onto the grid floor of the outcome box whereas 'retreats' were defined as partial entry into the outcome box followed by an immediate exit or backward treading into the neutral box." Why is there an asymmetry in the definition of entries and retreats? That is, the definition of entry is very clearly defined in terms of hind limbs making contact with the grid floor whereas the definition of retreat is more subjectively defined as "partial entry to the outcome box followed by an immediate exit or backward treading into the neutral box." What is a partial entry: i.e., what criteria were used to define a partial entry? Are 'retreats' capturing something that could be referred to as 'hesitancy'?

*Reviewer #2 (Recommendations for the authors):*

1) The authors should include histological reports of their implants for all optogenetic experiments.

2) There is no stated justification for the exclusion of female subjects. The authors should qualify their results indicating that they characterized the male rodent brain; this should be reflected in the title and abstract.

*Reviewer #3 (Recommendations for the authors):*

1. A major concern is the possibility that the perirhinal cortex is important for the identification of the conflicting stimuli, rather than in processing the conflicting motivation per se. This confound arises because optogenetic suppression occurs during the exposure to the objects. To address this concern, then, the authors should include an additional control for their first experimental paradigm in which they start suppression only after the entrance to the Neutral box or Goal box. If the perirhinal cortex is responsible for mediating motivational conflict, this should be sufficient to produce the originally observed effect. Although some of the experiments partially address this issue, as the authors argue in the discussion, the controls included don't address the possibility that encoding and retrieval specifically for the more complicated conflict stimuli. Given the importance of this concept to the overall message of the paper, this possibility requires additional direct interrogation.

2. Some of the results are confusing as presented and would benefit from additional analysis. In particular, the lack of difference in the number of entries between laser and non-laser for the first task along with a substantial difference in the number of retreats in Figure 2b is at least superficially confusing. I assume that this is across multiple trials (since otherwise, the number of retreats would be at most 1 less than the number of entries) but this should be clarified in the text and the number of trials should be included in the figure legend. A latency to retreat metric similar to the latency to enter would also be useful to include.

3. While optogenetic effects are observed for comparisons of the laser and non-laser trials across mice in the ArchT groups for the PRC suppression, it would be useful to assess this difference within mice, for instance by subtracting time-spent in goal box in laser and non-laser conditions. This would allow easier direct comparison between ArchT and GFP groups. Other metrics, such as the escape latency included in Figure 3f, could also be compared in this way. On a related note, the number of animals used appears to vary across experiments (for instance there are smaller number of dots in Figure 2c than other panels) but I do not see this noted in the figure legend or text. This should be clarified.

4. The difference in the Laser ON and Laser OFF for the example of c-fos staining with ArchT suppression of the entorhinal cortex shown in Figure 7d is not very convincing, which is surprising given the quantification in the previous panel. This should be addressed.

---

## [Author Response]

Essential revisions:Reviewer #1 (Recommendations for the authors):General1. Inhibition of the PRC altered conflict processing in the first and second tasks (runway and shuttle box, respectively) but not in the third (Y-maze). How do we know that the failure to detect an effect of PRC inhibition in the third conflict task is due to the fact that it requires context/spatial processing as opposed to reduced efficacy of the repeated PRC inhibition?

The reviewer has a valid concern regarding the use of repeated laser application. However, we are confident that the observed failure to detect an effect of PRC inhibition in our Y maze task is a meaningful outcome, based on the fact that we conducted another behavioural test (Novel Object Recognition task) *after* the Y-maze task, and observed that the laser application led to behavioural effects consistent with PRC inhibition, namely reduced novel object preference. In addition, subsequent immunohistochemical analysis revealed a significant reduction in cFos labeling in ArchT animals only following laser stimulation during this test.

2. How did animals interact with the object pairs in conflict trials? That is, did they preferentially explore one or the other object, and did PRC inhibition alter this exploration in a manner that correlates with its impact on test performance? I.e., did it create a bias towards the exploration of the object that signaled the appetitive outcome? The authors indicate that the exploration data were recorded; their inclusion in the manuscript could help to clarify how inhibition of the PRC has the effects that it does in the conflict test.

We did indeed record object exploration data and found that there were no differences between App and Av object exploration during any of the conflict tests. The object exploration data have now been included in the revised version of the manuscript (Supplementary Figure 2).

3a. L151. "When animals were exposed to a high conflict pairing, optogenetic inhibition of PRC (laser on) significantly increased time spent in the goal box compared with trials completed without inhibition (laser off) and compared to GFP control animals with laser on and off (laser x construct: F(1, 16) = 29.03, p < 0.001, ηp2 = 0.65; all post hoc: p < 0.001) (Figure 2a), indicative of an increase in approach behaviour under motivational conflict. Furthermore, PRC inhibition led to a significant decrease in the number of retreats from the goal box (laser x construct: F(1,16) = 16.80, p < 0.001, ηp2 = 0.51), although there was no effect on the number of goal box entries (laser x construct: F(1,16) = 1.92, p = 0.19, ηp2 = 0.11) (Figure 2b) during the choice period." Does the fact that PRC inhibited rats stay longer in the goal box indicate an increase in approach behaviour? These rats enter the goal box just as readily as the controls (entry and latency data) but stay longer when neither outcome occurs; so perhaps they are less sensitive to the absence of sucrose than the controls? The fact that PRC inhibition also increased time in the goal box on trials that included aversive-neutral object pairs (~L169) suggests that it may generally reduce avoidance of an aversive outcome. Is this a fair description of the conflict results? That is, PRC inhibition may have resulted in an appetitive response bias by reducing sensitivity to (or decreasing avoidance of) an aversive outcome…3b. L490. "Using a set of original object-based AA paradigms, we found that optogenetic inhibition of the rodent PRC, but not the vCA3, resulted in both a robust increase in approach behaviour and decrease in avoidance behaviour during motivational conflict elicited by the representation of discrete object pairs associated with unmatching affective values." What is the evidence of an increase in appetitive behaviour and decrease in avoidance behaviour?

The reviewer’s interpretation of our runway PRC inhibition data – that the increased approach bias is a result of decreased avoidance of the goal box is indeed a fair assessment, and one that cannot easily be disambiguated from the alternate account that PRC inhibition induces increased approach behaviour on the basis of our runway data. We believe we have evidence for both: the fact that we observed PRC inhibition-induced increase in the time spent in our ‘Av-Neu low conflict’ condition, in addition to our PRC inhibited rats exhibiting a prolonged ‘escape latency’ when first exposed to conflict object pairs and greater time spent in the conflict cued side in the shuttle box task appear to suggest that the decreased avoidance account may hold true. On the other hand, we also have strong evidence that PRC inhibited animals spend a disproportionate amount of time by the sucrose dispenser, seeking out reward under extinction conditions, which speaks to the presence of a robust increase in approach behaviour. We have inserted additional text in the Discussion to touch on this issue (revised manuscript page 30, para 2).

Design1. How were the rats allocated to groups for the Neu-Neu test shown in Figure 2c? Why was this testing conducted between-subjects (resulting in ns of just 5 per group) rather than within-subjects, as in the previous high conflict test? Did the 5 rats in each ArchT group (laser on vs laser off) and each GFP group (laser on vs laser off) exhibit the same profile of responding in the previous high conflict test? An obvious concern is that the allocation of rats to groups biased against detection of differences among the ArchT groups.

The reviewer’s point is well taken. Laser application during the Neu-Neu test, although not ideal, was conducted in a between-subjects design since we deemed that a fully counterbalanced experimental design with 2 conflict trials and 2 neutral trials might lead to unreliable data during the last trial owing to extinction-related confounds. We therefore prioritized maximizing the reliability of the data collected from the 2 conflict trials. The animals were allocated into the Laser on/off groups randomly, after it was confirmed that baseline performance during the conflict trials was comparable across animals (see revised manuscript, page 39, para 3).

2a. Why are there so few rats in the "Low Conflict-GFP" conditions? That is, why weren't rats in the "High Conflict-GFP" condition carried forward to these additional tests? It renders analyses of the Low and NO conflict conditions problematic: as there are 8 rats in the ArchT group and 4 rats in the GFP group, how can these be submitted to any sort of common analysis via ANOVA?2b. How many rats were in the ArchT and GFP groups for the novel recombination tests in the runway experiment? The figure again suggests 8 and 4, which again renders the ANOVA problematic.2c. The query in relation to unequal ns and application of ANOVA also applies to the results of the shuttle box experiment shown in Figure 3f-i (n for ArchT = 7; n for GFP = 4); and to the combination of PRC and vHPC data that is reported to have been analysed together near line 379. Can this be legitimized in some way?

We agree that there were issues with unequal group sizes here and in response, we have collected data from an additional 4 animals to add to our GFP group. Importantly, the overall pattern of findings remains unchanged from that reported in our original manuscript.

3. L640, 652, 664. How were rats tethered to the lasering in the AA conflict test as "the entire apparatus [was] wrapped in red cellophane to minimize reliance on extra-maze cues"?

We apologise that this detail was unclear. The top of the maze was exposed and uncovered to allow for access to the laser machine (see revised manuscript, page 34, para 2).

Data/stats1. L551: "An intriguing question is why PRC inhibition potentiated approach, as opposed to avoidance, of an uncertain outcome." I understand that there is no effect of the laser on latency to enter any of the runway sections in the high and low conflict tasks. I assume that this is due to the fact that latency data is notoriously variable and the ns are relatively low. As such, I would be interested in an assessment of whether there is an effect of the laser on latency to enter any of the runway sections in an analysis that combines high and low conflict tasks that include an appetitive component…

As per the reviewer’s suggestion, we conducted a 4-way ANOVA (Construct (ArchT: GFP) x Laser (on: off) x Conflict (Low: High) x Box (Objects: Neutral: Outcome)) of the latencies to enter (LTE) all the compartments of the runway across high vs. low conflict trials with an appetitive object. We found that ’laser on’ decreased entry times at both conflict levels, which was driven predominantly by the PRC inhibition group. We also observed that rats exhibited longer latencies to enter all compartments during high conflict trials compared to low conflict trials, which is consistent with the idea that a conflict object pair induced increased ambulatory/hesitant behaviour. These analyses have been included in the revised manuscript (page 11, para 2), and we believe that they strengthen our account that PRC inhibition potentiates reward approach.

2. In task 2, are the escape latency and "time in cued side" data correlated? If so, they are potentially two measures of the same thing, which would warrant some correction in the α rate to control the chance of a Type 1 error. This may have consequences for claims based on these data (e.g., analysis of data presented in Figure 3g).

Thank you for your comment regarding the correlation between escape latency and "time in cued side" data in task 2 and potential inflation of the α rate. “Escape latency” and “time in cued side” are indeed significantly correlated for both laser on (r=0.8) and laser off trials (r=0.62). However, while they may be measuring aspects of the same underlying construct, the measures are quite different, with the escape latency capturing the first ‘stay’ time in the cued side only, while the ‘time in cued side’ measure captures the time spent across multiple stays in the chamber (see revised manuscript, page 18, para 1).

Moreover, we would like to clarify that escape latency and "time in cued side" were analysed separately and thus, we do not believe that there is a concern about inflating the α rate or Type 1 errors.

3. L765. "The number of entries into and 'retreats' from the outcome box were recorded; an entry was defined when the animal's hind limbs stepped onto the grid floor of the outcome box whereas 'retreats' were defined as partial entry into the outcome box followed by an immediate exit or backward treading into the neutral box." Why is there an asymmetry in the definition of entries and retreats? That is, the definition of entry is very clearly defined in terms of hind limbs making contact with the grid floor whereas the definition of retreat is more subjectively defined as "partial entry to the outcome box followed by an immediate exit or backward treading into the neutral box." What is a partial entry: i.e., what criteria were used to define a partial entry? Are 'retreats' capturing something that could be referred to as 'hesitancy'?

Thank you for raising this point. Our usage of the term ‘retreat’ is consistent with how it was originally defined in our previously published approach-avoidance conflict work using a similar entry/retreat paradigm (e.g., Schumacher et al. 2018) and can indeed be regarded as an index of hesitancy. The definitions of a ‘partial entry’ and ‘retreat’ have been clarified in the revised manuscript on page 39, para 2.

Reviewer #2 (Recommendations for the authors):1) The authors should include histological reports of their implants for all optogenetic experiments.

Thank you for this comment. This was amiss on our part, and we have now added a brief report at the end of the Results section (page 27, para 3) as follows:

‘Finally, histological analysis confirmed robust bilateral expression of GFP/ArchT and optic fiber tip placement immediately dorsal to the viral injection site in all PRC animals. Thus, no exclusions were made based on optic fibre placement/viral expression. In the vCA3 group, data from three ArchT-expressing animals were excluded based on the viral expression and optic fibre placement presenting too medially to the CA3 subfield. ‘

2) There is no stated justification for the exclusion of female subjects. The authors should qualify their results indicating that they characterized the male rodent brain; this should be reflected in the title and abstract.

It is regrettable that we did not include female rats in the present study. The incorporation of females in our experiments has been a more recent commitment in our laboratory, and data collection for this study commenced 5 years ago. The title and abstract have been revised to reflect the use of an exclusively male rat population in the present study.

Reviewer #3 (Recommendations for the authors):1. A major concern is the possibility that the perirhinal cortex is important for the identification of the conflicting stimuli, rather than in processing the conflicting motivation per se. This confound arises because optogenetic suppression occurs during the exposure to the objects. To address this concern, then, the authors should include an additional control for their first experimental paradigm in which they start suppression only after the entrance to the Neutral box or Goal box. If the perirhinal cortex is responsible for mediating motivational conflict, this should be sufficient to produce the originally observed effect. Although some of the experiments partially address this issue, as the authors argue in the discussion, the controls included don't address the possibility that encoding and retrieval specifically for the more complicated conflict stimuli. Given the importance of this concept to the overall message of the paper, this possibility requires additional direct interrogation.

In order to address the reviewer’s concern, we have conducted a follow-up ‘control’ experiment in an additional cohort of animals (N=16; ArchT=8, GFP=8), with all rats completing both ‘High’ and ‘Low’ conflict tests in the runway paradigm, along with a ‘High’ conflict test in the shuttle box paradigm. When optogenetic inhibition was applied immediately prior to the decision phase of the task (i.e., following object exploration), the behavioural data (e.g., time spent in goal box; escape latency) did not differ from trials in which optogenetic inhibition was applied throughout the entire trial (i.e., data reported in our original manuscript). Furthermore, PRC-inhibited animals (ArchTLaser ON) in this control experiment also exhibited an increase in ‘reward approach’ behaviour by spending more time by the sucrose dispenser during ‘High’ conflict trials, underlining our suggestion that PRC is responsible for mediating object-based motivation conflict.

2. Some of the results are confusing as presented and would benefit from additional analysis. In particular, the lack of difference in the number of entries between laser and non-laser for the first task along with a substantial difference in the number of retreats in Figure 2b is at least superficially confusing. I assume that this is across multiple trials (since otherwise, the number of retreats would be at most 1 less than the number of entries) but this should be clarified in the text and the number of trials should be included in the figure legend. A latency to retreat metric similar to the latency to enter would also be useful to include.

We apologize that this was not clear in our original manuscript but given how we define the terms ‘entry’ and ‘retreat’, it is possible that the latter can occur independent of the former, and therefore, it is possible to have more retreats than entries. An entry occurred when the animal’s hind limbs stepped on to the grid floor of the goal box. To be consistent with our previous work using the same Y-maze task (Schumacher et al., 2018), a retreat occurred when the animal exhibited a half-body entry (e.g., only forelimbs) or head-poke into the goal box followed by an immediate exit or backward treading into the neutral box (see revised manuscript page 38, para 3).

3. While optogenetic effects are observed for comparisons of the laser and non-laser trials across mice in the ArchT groups for the PRC suppression, it would be useful to assess this difference within mice, for instance by subtracting time-spent in goal box in laser and non-laser conditions. This would allow easier direct comparison between ArchT and GFP groups. Other metrics, such as the escape latency included in Figure 3f, could also be compared in this way. On a related note, the number of animals used appears to vary across experiments (for instance there are smaller number of dots in Figure 2c than other panels) but I do not see this noted in the figure legend or text. This should be clarified.

We thank the reviewer for this suggestion. We have now included a difference score data analysis and presentation (Mean Data + SD from Laser on – Laser off) for a direct comparison of ArchT vs. GFP group data (see revised manuscript page 5, para 2; page 7, para 2; page 8, para 2; page 13 bottom; page 14, para 3; page 16 bottom; page 20, para 1) , alongside our original data. We decided to leave the complete data set in our manuscript for transparency.

We have also added more control GFP animals to the experiment and have clarified the Neu-Neu test design and methodology section, explaining the justification for a between-subjects test here (leading to an odd number of animals) (see revised manuscript page 39, para 3).

4. The difference in the Laser ON and Laser OFF for the example of c-fos staining with ArchT suppression of the entorhinal cortex shown in Figure 7d is not very convincing, which is surprising given the quantification in the previous panel. This should be addressed.

Higher magnification and resolution photomicrographs have now been included in the manuscript, providing a clearer illustration of the reduction in c-fos labelling following laser application in ArchTPRC tissue.